# Cbp1 and Cren7 form chromatin-like structures that ensure efficient transcription of long CRISPR arrays

Fabian Blombach [1] ✉, Michal Sýkora [1], Jo Case[1], Xu Feng [2], Diana P. Baquero [3], Thomas Fouqueau[1], Duy Khanh Phung[1], Declan Barker[1], Mart Krupovic [3], Qunxin She [2] & Finn Werner [1] ✉

CRISPR arrays form the physical memory of CRISPR adaptive immune systems by incorporating foreign DNA as spacers that are often AT-rich and derived from viruses. As promoter elements such as the TATA-box are AT-rich, CRISPR arrays are prone to harbouring cryptic promoters. Sulfolobales harbour extremely long CRISPR arrays spanning several kilobases, a feature that is accompanied by the CRISPR-specific transcription factor Cbp1. Aberrant Cbp1 expression modulates CRISPR array transcription, but the molecular mechanisms underlying this regulation are unknown. Here, we characterise the genome-wide Cbp1 binding at nucleotide resolution and characterise the binding motifs on distinct CRISPR arrays, as well as on unexpected non-canonical binding sites associated with transposons. Cbp1 recruits Cren7 forming together 'chimeric' chromatin-like structures at CRISPR arrays. We dissect Cbp1 function in vitro and in vivo and show that the third helix-turn-helix domain is responsible for Cren7 recruitment, and that Cbp1-Cren7 chromatinization plays a dual role in the transcription of CRISPR arrays. It suppresses spurious transcription from cryptic promoters within CRISPR arrays but enhances CRISPR RNA transcription directed from their cognate promoters in their leader region. Our results show that Cbp1-Cren7 chromatinization drives the productive expression of long CRISPR arrays.

Chromatinization of DNA regulates transcription in all domains of life, not only in eukaryotes, but also in bacteria[1] and archaea[2,3]. Chromatin proteins can regulate transcription by modulating the access of transcription factors and RNA polymerase (RNAP) to the gene promoter, but also suppress transcription from cryptic promoters inside genes which in particular in the antisense direction can be detrimental to productive gene expression.

In *Salmonella* and *Escherichia coli*, the chromatin protein H-NS preferentially binds to AT-rich DNA that is enriched in cryptic promoter elements including the Pribnow box, or '−10' element that is AT-

rich[4]. Deletion of H-NS in *E. coli* leads to recruitment of RNA polymerase to these cryptic promoters thereby depleting the pool of free RNA polymerase[5].

CRISPR-Cas systems evolved as adaptive immune system of prokaryotes against mobile genetic elements. The physical memory of this system is formed by CRISPR arrays, genomic regions that encompass clusters of repeat sequences interspersed by spacers derived from foreign DNA[6]. Transcription of these CRISPR arrays produces pre-crRNAs that are further processed into crRNA units encompassing single spacer-repeat units. Pre-crRNAs are amongst the longest non-

[1]RNAP laboratory, Institute for Structural and Molecular Biology, Division of Biosciences, University College London, Gower Street, London WC1E 6BT, United Kingdom. [2]CRISPR and Archaea Biology Research Center, Microbial Technology Institute, Shandong University, Qingdao 266237, PR China. [3]Institut Pasteur, Université Paris Cité, CNRS UMR6047, Archaeal Virology Unit, F-75015 Paris, France. ✉e-mail: f.blombach@ucl.ac.uk; f.werner@ucl.ac.uk

coding RNAs in prokaryotes next to the 16S and 23S rRNAs. In *E. coli*, dedicated antitermination complexes ensure processivity of CRISPR array transcription by preventing premature Rho-dependent transcription termination similar to the antitermination complexes facilitating rRNA transcription[7]. Beyond *E. coli*, little is known about specific mechanisms controlling CRISPR array transcription. Sulfolobales harbour a number of general chromatin proteins, in particular Alba, Sul7 and Cren7[2]. Initial chromatin immunoprecipitation sequencing (ChIP-seq) data suggested increased occupancy of Cren7 at CRISPR arrays in *Saccharolobus solfataricus*, but whether this affects CRISPR function remains unknown[8].

Members of the Sulfolobales order have multiple CRISPR systems and multiple CRISPR arrays spanning often >100 spacers and resulting in pre-crRNAs of several kilobases in length. The extraordinary length of CRISPR arrays in Sulfolobales coincides with the presence of the CRISPR array binding protein 1, Cbp1, that binds to the repeat sequences in CRISPR arrays[9,10]. Cbp1 comprises three helix-turn-helix (HTH) domains that are derived from domain duplications[9,10].

Deletion of *cbp1* in *Saccharolobus islandicus* leads to reduced levels of pre-crRNA and processing intermediates, while Cbp1 overexpression increases the levels of pre-crRNA[9]. These observations lead to the concept that Cbp1 is a positive transcription elongation factor specific for CRISPR arrays[9], which is counterintuitive as a protein binding to DNA at numerous sites downstream of an elongating RNAP likely acts as a 'roadblock' factor. An alternative, apparently mutually exclusive hypothesis suggests that the regulated binding of Cbp1 to CRISPR arrays might induce premature transcription termination to adjust the relative levels of different crRNAs[11]. New spacers that provide greater protection against the current viral pool are generally integrated at the 5′ end of CRISPR arrays[12–15]. Premature transcription termination enriches the crRNAs bearing new spacers in the total crRNA pool and counteract the 'dilution effect' arising from the transcription of distal spacers in very long CRISPR arrays[11].

Previous data suggested that Cbp1 suppresses spurious transcription from internal promoters[9]. Cbp1 could also have additional functions, e.g. it could facilitate the regulation of CRISPR systems in response to infection, e.g. during spacer acquisition. Infection with the SIRV2[16] and STSV2[17] viruses induces upregulation of CRISPR array expression in *Saccharolobus islandicus*, but it remains unknown whether this coincides with altered Cbp1 chromatinization of CRISPR arrays.

Here we provide a genome-wide yet detailed characterisation of Cbp1 function in CRISPR array transcription that enhances our understanding of the regulation of CRISPR systems and provides more general insights into the role of unorthodox chromatin-like proteins in transcription regulation in prokaryotes. We have mapped Cbp1 binding patterns globally and at nucleotide resolution, dissected the binding motifs and modes on distinct arrays, and tested the impact of Cbp1 binding on transcription in vitro and in vivo. We show that Cbp1 directly recruits Cren7 to the 3′-end of CRISPR repeats suppressing transcription initiation from cryptic promoters in spacers while facilitating transcription from CRISPR leader promoters. Additional binding sites of Cbp1 associated with transposases and the leaders of alternative CRISPR arrays hint on a wider regulatory function of Cbp1 linking defense systems and mobile genetic elements.

## Results

### Cbp1 recruits Cren7 to chromatinize CRISPR arrays

To gain insight into the chromatinization of CRISPR arrays in vivo and to test whether Cbp1 and Cren7 chromatinization of CRISPR arrays are interdependent, we determined the genome-wide occupancy of Cbp1 by ChIP-seq in three *Saccharolobus* strains: (i) *S. solfataricus* P2 that harbours six CRISPR arrays (A to F) with slight differences in their repeat sequences, (ii) *S. islandicus* LAL14/1 that harbours three CRISPR arrays with different, unrelated repeat sequences alongside two

CRISPR arrays from the same family of repeats as those found in *S. solfataricus*[18], and (iii) the genetically tractable *S. islandicus* REY15A allowing us to test the effect of *cpb1* deletion on Cren7 occupancy. *S. islandicus* and *S. solfataricus* are closely evolutionary related and the two Cbp1 (P2 vs. REY15A and LAL14/1) proteins share 93% amino acid identity.

All six CRISPR arrays in *S. solfataricus* P2 showed increased ChIP-seq occupancy for both Cbp1 and Cren7 (Fig. 1a). The six CRISPR arrays can be classified into three groups based on their CRISPR repeat sequences with A/B and C/D having identical repeats, respectively, while E/F repeats differ by a single nucleotide (Fig. 1b). Notably, Cbp1 and Cren7 occupancy was correlated between the groups of CRISPR arrays (noticeable as distinct clusters Fig. 1c) as well as within each cluster formed by each CRISPR array group (Fig. 1c).

In addition to the CRISPR array associated binding of Cbp1, we also identified several non-canonical binding sites for Cbp1 in *S. solfataricus* P2 (92 peaks with at least five-fold enrichment). These were frequently associated with intact or partial *ISC1229* transposons (an example is shown in Supplementary Fig. 1). These binding sites featured a 21 bp motif with a consensus sequence matching the CRISPR repeat consensus sequence with the strongest sequence conservation at six base pairs close to the 3′-end of the repeats (Fig. 1b). Notably, these non-canonical Cbp1 binding sites in *S. solfataricus* did generally not show any enrichment of Cren7 compared to the genomic background (Fig. 1c).

In *S. islandicus* LAL14/1, CRISPR arrays 1 and 2 have repeat sequences identical to CRISPR F in *S. solfataricus* P2 and Cbp1 ChIP-seq showed that they are bound by Cbp1 (Supplementary Fig. 2). In contrast, arrays 3, 4 and 5 bearing an unrelated repeat sequence were not bound by Cbp1 (Supplementary Fig. 2). The CRISPR repeat sequences display differential prevalence in Sulfolobales species from different geographic locations, with the array 1/2-like (F-type) repeat sequence being dominant in most analysed locations[19].

To test whether there is a causal relationship between Cbp1 and Cren7 chromatinization of CRISPR arrays, we tested Cren7 occupancy on CRISPR arrays in a *cbp1* deletion strain generated in *S. islandicus* REY15A strain E233S,[9]. Cbp1 and Cren7 chromatinized the two F-type CRISPR arrays in REY15A. Cren7 expression levels are not affected by *cbp1* deletion (Supplementary Fig. 3), but Cren7 occupancy on the two CRISPR arrays was severely reduced (Fig. 1d). Deletion of Cbp1 is accompanied by a deletion of a ~28 kb genomic region between two IS*200*/IS*605* family transposases, SiRe_0633 (SIRE_RS03230) and SiRe0665 (SRE_RS03390). However, this deletion is unlikely to affect Cren7 binding (see Supplementary Table 1).

In summary, our ChIP-seq data thus suggest that Cbp1 enhances or facilitates Cren7 recruitment. We tested this hypothesis directly in electrophoretic mobility shift assays (EMSA) on dsDNA templates with a single CRISPR repeat from CRISPR arrays A, D and F or a non-canonical binding site derived from an IS*C1229* transposon. The EMSAs confirmed Cbp1-dependent recruitment of Cren7 to all DNA templates. Notably, Cbp1-bound CRISPR A repeats appeared to have an overall higher binding affinity for Cren7 and a second Cren7 monomer was recruited to Cbp1-bound CRISPR A repeats at higher Cren7 concentrations (Fig. 1e). Cbp1 itself appeared to show weaker affinity for the CRISPR A repeat than for the CRISPR F repeat in direct contrast to our ChIP-seq occupancy data but in line with previous findings[9]. The Cren7 concentrations used in our experiments were below the range where Cren7 efficiently chromatinizes DNA[20] explaining the absence of detectable DNA-binding by Cren7 in the absence of Cbp1.

Recruitment of Cren7 could be mediated by direct physical interaction between Cbp1 and Cren7 or by Cbp1-induced topological changes in the DNA template facilitating Cren7 binding. To test whether there is a direct physical interaction, we conducted cross-linking assays using the amine-specific cross-linker Bis(sulfosuccinimidyl) suberate (BS³). We observed a cross-linked species corresponding to

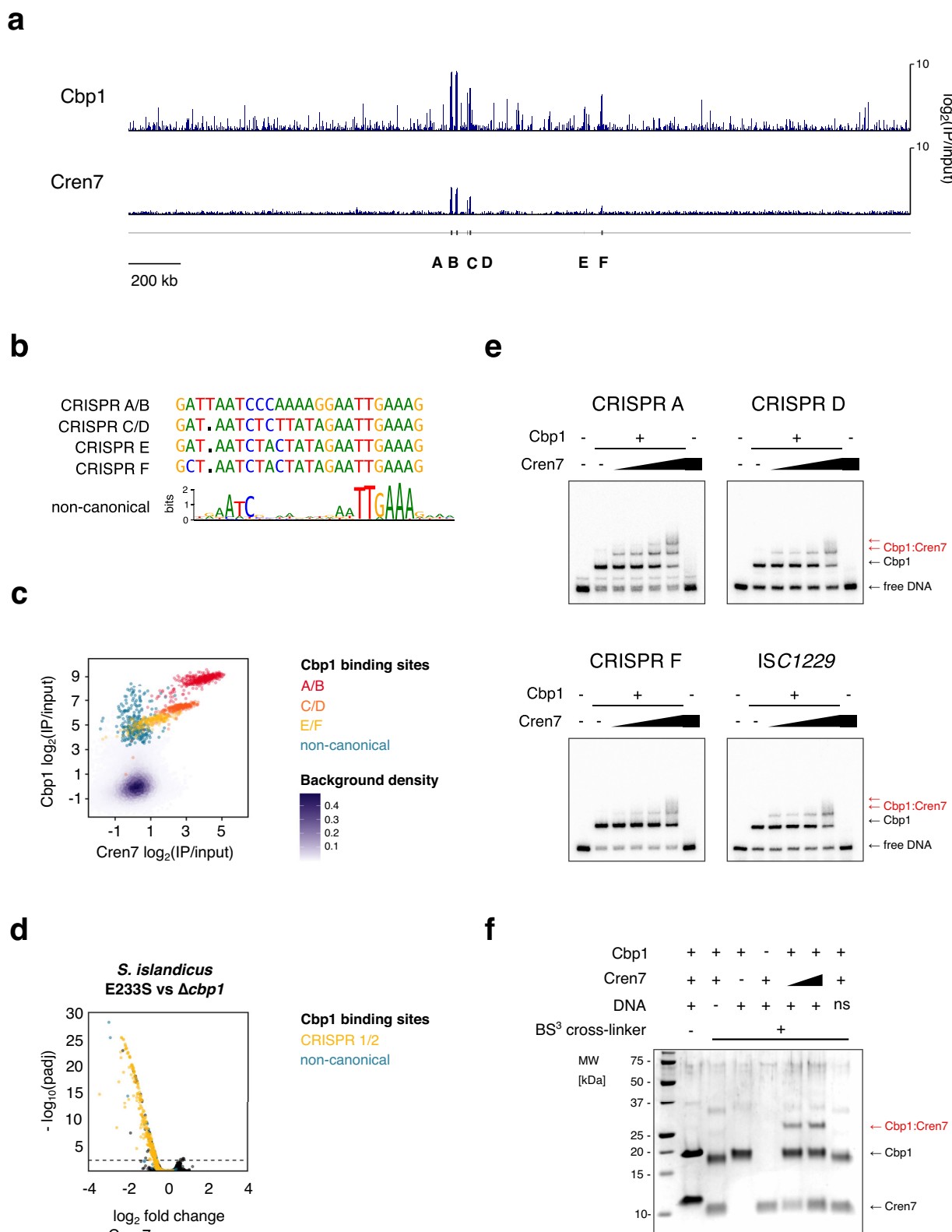

the expected combined molecular weight of Cbp1 and Cren7 dependent on the presence of Cbp1, Cren7, and a DNA template bearing a CRISPR repeat (Fig. 1f). The cross-linked species was not observed in control experiments with a non-specific DNA template (Fig. 1f). Our data thus demonstrate that Cbp1 recruits Cren7 through a combination of direct physical interactions and DNA binding.

We estimated the transcript abundance of *cbp1* and *cren7* in *S. solfataricus* P2 and *S. islandicus* REY15A (E233S) based on RNA-seq data. Transcript levels for *cbp1* were between 5 to 8% of *cren7* transcript levels (Supplementary Table 2). Taking into account the small fraction of the genome bound by Cbp1 compared to Cren7, a highly abundant general chromatin protein[20], the data suggest that Cbp1 is

**Fig. 1 | Cbp1 recruits Cren7 to chromatinise CRISPR arrays. a** Genome-wide overview of Cbp1 and Cren7 occupancy on the *S. solfataricus* P2 genome. The location of the six CRISPR arrays is indicated. ChIP-seq data represent the mean of two biological replicates with 2000 bp bin size. **b** Sequence alignment of the different repeat sequences in the *S. solfataricus* P2 genome and WebLogo of the motif identified in non-canonical Cbp1 binding sites (n = 92). **c** Scatter plot depicting correlated occupancy of Cbp1 and Cren7 on CRISPR arrays. Average occupancy (normalised against input) for two biological replicates was calculated over 30 bp consecutive bins. **d** Cren7 recruitment to CRISPR arrays is reduced in a *cbp1* deletion strain. Volcano plot showing the differential binding analysis of Cren7 ChIP-seq data for *S. islandicus* E233S (WT) and Δ*cbp1*. Read counts were calculated for 30 bp consecutive bins for two biological replicates. **e** Cbp1 facilitates recruitment of Cren7. EMSA testing Cbp1 and Cren7 binding to three different CRISPR repeats and a non-canonical binding site of Cbp1. 12.5 nM Cbp1 and 3.125 to 25 nM Cren7 (2x dilution series) were incubated with 5′-radiolabelled

dsDNA template encompassing the CRISPR repeats and 20 bp flanking spacer DNA on either side or the corresponding region for a non-canonical binding site derived from a IS*C1229* transposon. Representative gels of three technical replicates are shown. **f** Cbp1 and Cren7 directly interact with each other in a DNA-dependent manner. Cbp1:Cren7:CRISPR A repeat DNA complexes were assembled with 2.5 μM Cbp1, 2.5 or 5 μM Cren7, and 2.5 μM dsDNA template. The complexes were then subjected to protein:protein cross-linking with 1 mM BS$^3$. Cross-linked samples were resolved by SDS-PAGE with Coomassie staining. A Cbp1:Cren7 cross-linked species of ~25 kDa (labelled in red) was formed strictly in the presence of BS$^3$, Cbp1, Cren7 and the CRISPR A repeat 1 template. Replacement of the CRISPR DNA template with a non-specific control DNA did not yield any detectable Cbp1:Cren7 cross-linked species. Some background signal at ~37 kDa can be attributed to a small fraction of Cbp1 dimers. A representative gel from three replicate experiments is shown.

abundant enough to reach high levels of chromatinization at strong binding sites such as CRISPR arrays A and B.

## The essential nature of Cren7 does not depend on Cbp1

*Cren7* is essential for *S. islandicus* viability, but *cbp1* is not[9,21]. Deletion of *cbp1* coincides with loss of Cren7 binding to its main target in the genome, the CRISPR arrays. To test a genetic interaction between the two chromatin proteins, we attempted to delete *cren7* (SiRe_1111) in the Δ*cbp1* strain. However, we were unable to delete *cren7* in *S. islandicus* Δ*cbp1* whereas the non-essential gene SiRe_0782[21] that serves as positive control could be deleted (Supplementary Table 3). We conclude that the lethality of the *cren7* deletion cannot be compensated for by concomitant deletion of *cbp1*, which indicates that the essentiality of *cren7* is not connected to CRISPR array chromatinization.

## Architecture of the Cbp1-Cren7-DNA complex

To gain insight into the topology of the Cbp1-Cren7-DNA complex, we used ChIP-exo, a method combining ChIP-seq with 5′ → 3′ exonuclease trimming, to map the foot-prints of Cbp1 and Cren7 on CRISPR repeats at nucleotide resolution. Aggregate profiles for Cbp1 foot-prints on CRISPR repeats of arrays A and B revealed a 5′ border at position −5 relative to the start of the CRISPR repeat and a 3′ border located at position +16 downstream of the repeat start (Fig. 2a). Notably, position +16 is upstream of the region encompassing the core Cbp1 binding motif (Fig. 1b). Cbp1 foot-prints for the non-canonical Cbp1 binding sites showed a similar pattern with the 3′ border located at the corresponding position immediately upstream of the 3′-terminal core binding motif (Supplementary Fig. 4). Because of this lack of protection over the core binding motif, the Cbp1 ChIP-exo foot-prints appear not to represent the full extent of the Cbp1 binding site and are potentially biased by the cross-linkability at different positions. The corresponding ChIP-exo profiles for Cren7 were highly similar to those obtained for Cbp1 potentially due to stronger protein-protein cross-linking relative to DNA-protein cross-linking at our experimental conditions consistent with other ChIP-exo data we obtained previously[22]. Nevertheless, the Cren7 aggregate profile showed additional protection downstream of position +16 downstream reaching into the 3′-flanking spacer sequence. Combined with the pattern of sequence conservation (Fig. 1b), the ChIP-exo data suggest that Cbp1:Cren7 binding covers the entire CRISPR repeat and stretches downstream into the flanking spacer.

To test the role of the downstream DNA flanking the core motif in Cbp1 and Cren7 binding in vitro, we conducted EMSAs with DNA templates including varying lengths of 3′-flanking spacer sequences. We observed recruitment of two Cren7 monomers was retained when the 3′-flanking DNA was trimmed from 20 bp to 2 bp (Fig. 2b). Further trimming of the remaining flanking DNA abolished Cren7 recruitment in line with Cren7 interacting with the 3′ flanking DNA (Fig. 2b).

## Dissection of Cbp1 HTH domain contributions to DNA binding

Cbp1 is composed of three HTH domains connected by short linker regions, these domains likely arose by domain duplication. To determine how the HTH domains contribute to CRISPR repeat binding, we deleted either the first or the third HTH domain (ΔHTH1 and ΔHTH3). Both deletion variants were expressed at comparable levels to the wild type full length protein, and their thermostability validated that the mutations had not impaired correct protein folding (Supplementary Fig. 5). We tested the relative orientation of the Cbp1 HTH domains on the CRISPR repeats by combining mutations in the CRISPR repeat with the Cbp1 ΔHTH1 and ΔHTH3 variants. If the N- or C-terminal HTH domain interacts with a specific region in the CRISPR repeat, we reasoned that mutations in this region should not affect binding of a Cbp1 variant where the HTH is absent. We tested four different double transversion mutations in the CRISPR repeat at positions corresponding to positions with strong sequence conservation in the Cbp1 binding motif: A6C/T7G, A17C/A18C, T19G/T20G, A22C/A23C. As a control, we also included the A13C/T14G mutation, a region without any apparent sequence conservation in the core motif derived from non-canonical binding sites (Fig. 1b). We first tested the effect of these mutations on full-length Cbp1. All mutations but the A13C/T14G control reduced Cbp1 binding relative to the WT repeat sequence (Fig. 2c, likelihood ratio tests of nested beta regression models with Bonferroni multiple testing correction, adjusted *p*-value < 0.001 for all mutations). Both Cbp1ΔHTH1 and Cbp1ΔHTH3 showed overall weaker binding. Crucially, Compared to WT Cbp1, Cbp1ΔHTH1 appeared to be unaffected by the 5′-terminal A6C/T7G mutation (likelihood ratio tests of nested beta regression models with Bonferroni multiple testing correction, adjusted *p*-value < 0.001) while retaining sensitivity to mutations in the 3′-end of the CRISPR repeat. Conversely, compared to WT Cbp1, Cbp1ΔHTH3 appeared to be generally less affected by mutations (possibly to some loss of binding specificity), including the 3′-terminal two mutations that drastically reduce binding of WT Cbp1 but have no apparent effect on Cbp1ΔHTH3 binding (adjusted *p*-value < 0.001, Fig. 2c). Our data suggest that the three HTH domains of Cbp1 align along the CRISPR repeat in 5′ to 3′ direction. We corroborated these results by testing Cren7 recruitment by the HTH-deletion variants in EMSAs. In line with HTH3 binding to the 3′-terminal core motif where Cren7 is recruited, the Cbp1ΔHTH3 mutant failed to recruit Cren7 (Fig. 2d).

We corroborated our data by carrying out DNase I foot-printing experiments with Cbp1 and Cren7. Previous DNase I foot-printing experiments for Cbp1 binding to CRIPSR repeats revealed a DNase hypersensitivity site in the centre of the repeat flanked by protected regions[10]. This hypersensitivity site is close to the main 3′ border signal we identified in our Cbp1 ChIP-exo foot-prints at position +16 of the repeat. Our DNase foot-printing data reproduced these findings (Fig. 2e). The addition of Cren7 enhanced the protection and extended it at least 6 bp into the spacer downstream of the repeat. A new minor

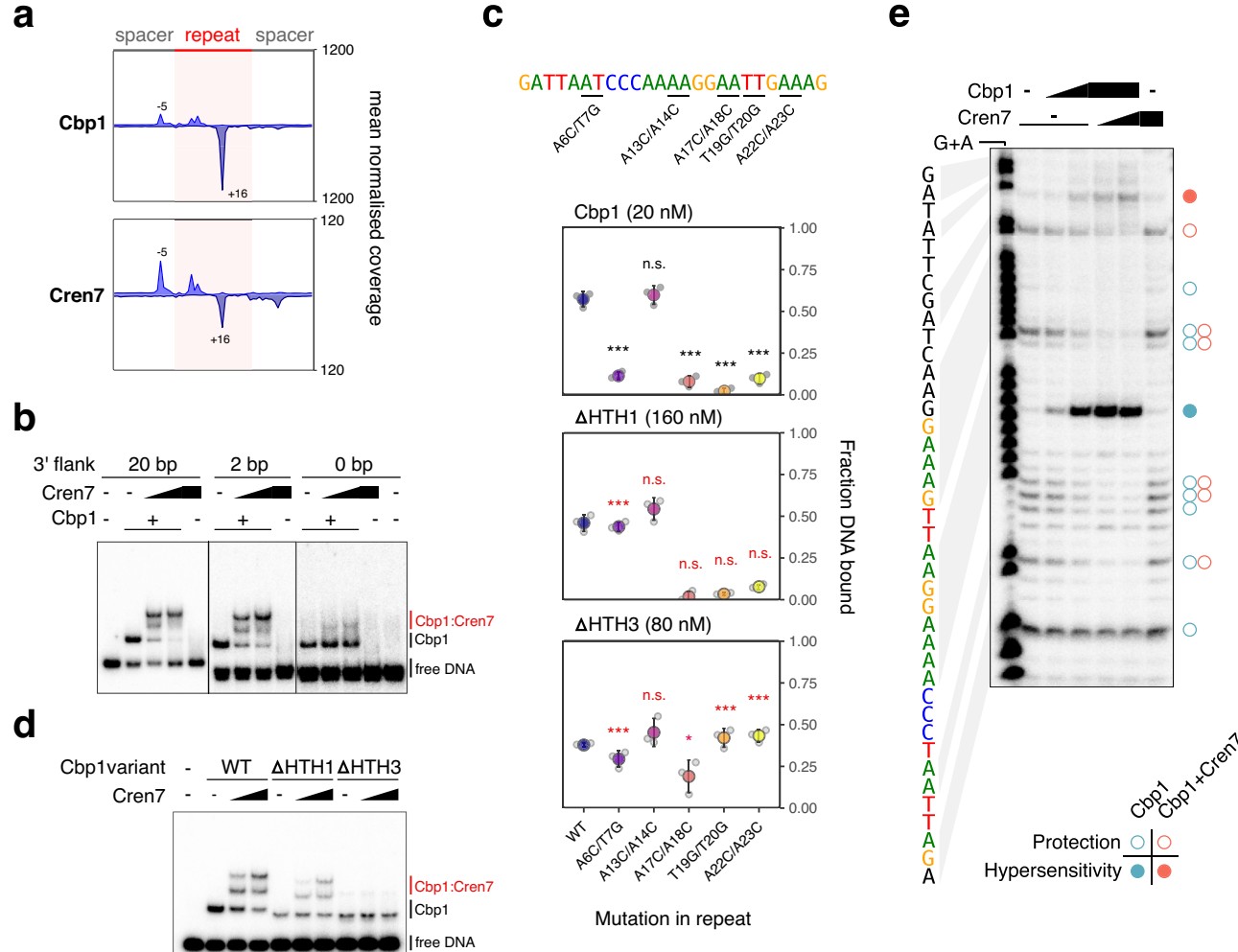

**Fig. 2 | Architecture of Cbp1:Cren7 chromatinization. a** Aggregate plots of Cbp1 and Cren7 ChIP-exo occupancy over 208 repeats from CRISPR arrays A and B. Signal for the plus and minus strand (relative to CRISPR array orientation) is plotted above and below the x-axis, respectively. The aggregate mean signal was calculated from the geometric mean of two biological replicates scaled to reads per million. **b** Cren7 recruitment to Cbp1:CRISPR DNA complexes depends on the flanking DNA downstream of the CRISPR repeat. EMSAs with templates bearing a CRISPR A repeat with 20 bp upstream and 20, 2 or 0 bp downstream flanking sequence and 12.5 nM Cbp1, 12.5 to 25 nM Cren7. The free DNA of the shorter templates appeared as double-band, possibly due to some denaturation at the incubation temperature of 75 °C. A representative gel from three replicate experiments is shown. **c** Orientation of Cbp1 binding to CRISPR repeats. Binding of Cbp1 and HTH-deletion mutants ΔHTH1 and ΔHTH3 to mutated CRISPR repeat sequences was assessed by EMSAs and the fraction of DNA bound by Cbp1 was plotted (mean of three biological replicates and standard deviation are shown). Double mutations were introduced into the CRISPR repeat sequence based on regions showing stronger sequence bias in the motif identified for non-canonical Cbp1 binding sites (Fig. 1b). The A13C/T14G double mutant was included as control

for a region of CRISPR repeats that shows no sequence bias in non-canonical binding sites. To compensate for the overall lower affinity of the ΔHTH1 and ΔHTH3 variants, their concentration was raised to 160 and 80 nM, respectively. The effect of CRISPR repeat mutations on WT Cbp1 binding (black asterisks) and whether these effects are altered with HTH deletions in Cbp1 (red asterisks) was tested in likelihood ratio tests of nested beta regression models with Bonferroni multiple testing correction (see Methods section for details). *** and * denote *p* adj <0.001 and 0.01, respectively. **d** Cren7 recruitment to Cbp1:CRISPR repeat complexes depends on the HTH3 domain of Cbp1. EMSA assay testing Cren7 recruitment (12.5 to 25 nM) to CRISPR DNA-bound Cbp1 (12.5 nM), ΔHTH1 (50 nM) and ΔHTH3 (25 nM). Representative gels from three replicate experiments are shown. **e** DNase foot-printing assays corroborate Cbp1-dependent Cren7 deposition at the downstream spacer. Foot-printing assays were carried out with 12.5 or 25 nM Cbp1 and 25 or 50 nM Cren7. A G + A sequencing ladder is shown on the left as reference. Cbp1 and Cbp1:Cren7-induced protection against DNase cleavage is indicated with open circles and hypersensitivity is indicated with full circles. A representative gel from three replicate experiments is shown.

hypersensitivity site appeared at position 10 in the downstream spacer (Fig. 2e). Taken together, our data establish the overall topology of the Cbp1-Cren7-DNA complex with Cren7 being recruited to the 3′-terminal region of the CRISPR repeat covering in part the flanking spacer.

**Cbp1-Cren7 chromatinization of CRISPR arrays suppresses spurious transcription from cryptic promoters**
Having established the topology of Cbp1-Cren7-DNA complex, we set out to investigate how Cbp1-Cren7 chromatinization affects transcription of CRISPR arrays. We compared transcription of

*S. solfataricus* P2 CRISPR B and CRISPR F that show high and low levels of Cbp1-Cren7 chromatinization, respectively. To probe how Cbp1-Cren7 chromatinization relates to the recruitment of the transcription machinery, we mapped the ChIP-seq occupancy of transcription initiation and elongation factors, as well as RNAP and regulatory factors using our previously published ChIP-seq data[22]. To complement these binding data with transcription output, we analysed short RNA sequencing data generated by Cappable-seq[22], a method that is highly selective for 5′-triphosphorylated RNA and thus capable to detect transcription initiation events with high sensitivity[23].

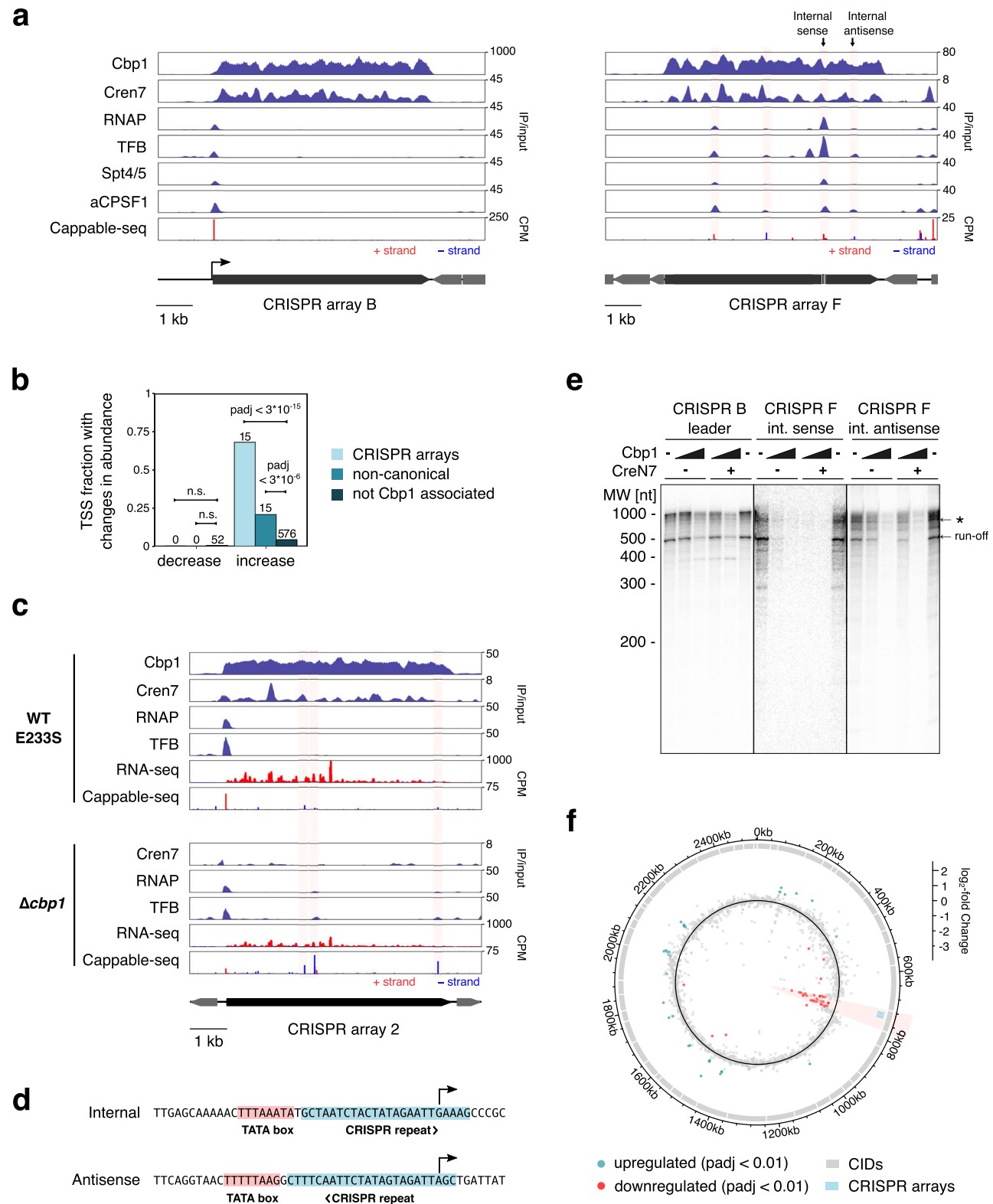

Our data revealed several TSSs associated with internal promoters both in sense and antisense orientation within CRISPR array F (Fig. 3a). Basal transcription factors including TBP, TFB and TFE form transcription preinitiation complexes with RNAP and facilitate promoter-directed transcription initiation. During early elongation, proximal to the promoter, the elongation factors Spt4/5 and Elf1, and the termination factor aCPSF1, are recruited to the transcription elongation complex[22,24]. The heterogenous factor composition and occupancy

detected at the internal promoters reflect the RNAP in different stages of the transcription cycle. E. g. the promoters in array B showed ChIP-seq occupancy of TFB and RNAP, but not Spt4/5, suggesting that these promoters allow only for preinitiation complex formation (Fig. 3a). In contrast, several promoters within array F showed occupancy of TFB, RNAP, Spt4/5 and aCPSF1 suggesting that RNAP had progressed to the transcription elongation complex and possibly even premature transcription termination. The ChIP-seq occupancy corresponded overall

**Fig. 3 | Cbp1:Cren7 chromatinization of CRISPR arrays prevents spurious transcription from internal promoters. a** *S. solfataricus* CRISPR arrays B and F with high and low Cbp1:Cren7 occupancy, respectively, show different levels of spurious transcription. ChIP-seq occupancy profiles for Cbp1, Cren7, RNAP, initiation factor TFB, elongation factor Spt4/5 and termination factor aCPSF1 are shown (mean of two biological replicates). The lower panel shows Cappable-seq data as 5′-end coverage for 5′-triphosphorylated short RNAs (~20–200 nt). The geometric mean from two biological replicates was scaled to counts per million (CPM). The position of the F1 internal and antisense promoters tested in in vitro transcription (panel **d**) are indicated. **b** Deletion of *cbp1* leads to activation of TSSs within CRISPR arrays and close to non-canonical binding sites of Cbp1. TSSs were mapped using Cappable-seq for *S. islandicus* REY15A E233S (2 biological replicates) and *cbp1* deletion strains 2 and 3. Differential expression of TSSs (*P*adj < 0.01) was identified using Deseq2. Enrichment of differentially expressed TSSs close to Cbp1 binding sites was assessed using Fisher's exact test (two-sided) with Bonferroni multiple testing correction. **c** *cbp1* deletion elevates spurious transcription from *S. islandicus* REY15A CRISPR arrays 1 and 2. ChIP-seq occupancy profiles for Cbp1, Cren7, RNAP, and TFB alongside RNA-seq and Cappable-seq data for the 5′-end coverage of triphosphorylated short RNAs are shown for parental strain E233S and the *cbp1* deletion strains[9]. The arithmetic mean coverage of two biological replicates is shown for all ChIP-seq (*cbp1* deletion strain 1) and RNA-seq data (*cbp1*

deletion strains 2 and 3). Cappable-seq data (*cbp1* deletion strains 2 and 3) represent the geometric mean. RNA-seq and Cappable-seq data were scaled to counts per million (cpm) for fragments and reads, respectively. **d** Sequence of the internal sense and antisense promoters in CRISPR F1 (see panel **a**) used in cell-free transcription experiments. **e** Cbp1 and Cbp1:Cren7 chromatinization prevents spurious transcription from CRISPR-internal promoters in vitro. The internal sense and antisense promoters in CRISPR F (see panels **a** and **d**) were tested alongside the CRISPR B leader promoter in cell-free transcription assays with increasing concentrations of recombinant Cbp1 added (0, 100, 300 nM) and in presence (300 nM) or absence of recombinant Cren7. Affinity-purified radiolabelled transcripts were resolved on a denaturing polyacrylamide gel. A representative gel of three technical replicates is shown. The expected run-off transcript size was 516 nt for CRISPR B leader and 505 nt for the two CRISPR F internal promoters. **f** *cpb1* deletion widely affects gene expression in the CID encompassing the two CRISPR arrays in *S. islandicus* REY15A. Circos plot showing $\log_2$-fold changes in relative RNA abundance for 2655 coding genes with >0 coverage and the two CRISRPR arrays from RNA-seq data for strain E233S (two biological replicates) versus *cbp1* deletion strains 2 and 3. Significantly up- and downregulated genes (*P*adj < 0.01) are highlighted in green and red, respectively. The ranges of 47 CIDs previously identified by Takemara and Bell[27] are indicated with the CID encompassing the CRISPR arrays highlighted.

well with the detection of 5′-triphosphorylated RNAs by Cappable-seq at these promoters. The internal cryptic promoters generally coincided with local minima in Cren7 ChIP-seq occupancy. These data suggest that Cbp1-Cren7 chromatinization might compete with transcription from cryptic promoters within CRISPR arrays.

To establish a causal relationship between Cbp1 chromatinization and the suppression of spurious transcription, we compared transcription profiles for *S. islandicus* Δ*cbp1* strains with the parental strain E233S by Cappable-seq and standard RNA sequencing as well as ChIP-seq for RNAP and TFB.

As in *S. solfataricus*, cryptic promoters appeared to be active in spacers with locally decreased Cren7 occupancy in E233S. To map changes in transcription activity of these cryptic promoters, we conducted a differential expression analysis of the Cappable-seq data for the identified TSSs in the *S. islandicus cbp1* deletion versus the parental strain. To avoid the additional deletion of a genomic region in our original Δ*cbp1* strain, we reconstructed the strain. The two biological replicates for the parental E233S strain as well as two independent Δ*cbp1* strains (strains 2 and 3) correlated well across the quantified signal for 13,150 TSSs (Spearman's r = 0.97 for the E233S replicates and r = 0.96 for the two Δ*cbp1* strains, Supplementary Fig. 6). The Cappable-seq data allowed us to identify 13,150 TSSs including 946 primary TSSs for 1506 predicted operons with ~60% of transcripts predicted to be leaderless (5′-UTR length <4 nt, Supplementary Fig. 7) similar to data for *S. solfataricus* P2[25]. The majority of TSSs were internal sense or antisense TSSs (6459 and 4052, respectively) again reflecting findings for *S. solfataricus* P2[25].

The differential expression analysis for the Cappable-seq data confirmed our hypothesis as the *cbp1* deletion results in substantial changes in TSS utilisation and transcriptome changes with 658 out of 13,150 TSSs differentially transcribed (*P*adj < 0.01, Fig. 3b). The TSSs associated with leader promoters that direct CRISPR array transcription appeared to be downregulated by the deletion of *cbp1* ($\log_2$-fold change of −1.56, *P*adj < 0.053 for CRISPR1 and −1.7, *P*adj < 1*10⁻¹⁵ for CRISPR2) in good agreement with the reduced level of pre-crRNA observed in Δ*cbp1* using Northern blots[9]. This effect could be caused by a reduced transcription initiation frequency or by a feedback mechanism from slow/paused transcription elongation complex interfering with transcription preinitiation complexes and productive initiation. Next, we investigated the effect of the Cbp1 deletion on cryptic promoters inside CRISPR arrays. A large fraction of TSSs and promoters residing within the two CRISPR arrays were transcribed at higher levels upon *cbp1* deletion (15 out of 22, *P*adj < 3*10⁻¹⁵, Fisher's exact test (two-sided), Bonferroni correction), in good agreement with

the hypothesis that Cbp1 binding suppresses internal and spacer promoter transcription. Non-canonical binding sites of Cbp1 that are not associated with CRISPR arrays featured a repeat-like binding motif consistent with *S. solfataricus* (Supplementary Fig. 8). TSSs associated with non-canonical Cbp1 binding sites, i.e. within 50 bp of Cbp1 peak summit, also showed a slightly higher fraction with increased expression after *cbp1* deletion (15 out of 72, *P*adj < 3*10⁻⁶). ChIP-seq data for RNA polymerase and TFB in the E233S WT and the Δ*cbp1* strain demonstrate that the increased transcription resulted from cryptic promoters active already in the WT strain. Thus, spurious transcription from CRISPR-array internal promoters appears to be suppressed by Cbp1:Cren7 chromatinization (Fig. 3c).

We previously described that global mRNA levels in archaea are correlated with the fraction of RNAPs that successfully 'escape' into the productive elongation phase of transcription. During normal growth, RNAP is predominantly located in the promoter-proximal regions of CRISPR arrays (Fig. 3a)[22]; the release from this state, an increase of RNAP escape, likely triggers the observed up-regulation of CRISPR expression in response to viral infection[16,17]. Notably, promoter-proximal peaks of RNA polymerase were also observed in the Δ*cbp1* strain indicating that RNA polymerase accumulation is not strongly dependent on Cbp1 chromatinization (Fig. 3c).

ChIP-seq results for RNAP occupancy on CRISPR arrays in LAL14/1 were consistent with these observations (Supplementary Fig. 2). The non-Cbp1-bound CRISPR arrays 3, 4, and 5 show internal RNAP peaks indicative of RNAP recruitment to cryptic internal promoters, whereas Cbp1-bound CRISPR arrays 1 and 2 show no signs of any RNAP peaks within the array.

To test the competition between Cbp1 binding and transcription directly, we used cell-free in vitro transcription assays comparing two internal promoters within *S. solfataricus* CRISPR array F (Fig. 3a), one in sense and one in antisense orientation, with the leader promoter of CRISPR array B that directs pre-crRNA transcription. The internal sense and antisense promoters are both overlapping with CRISPR repeats that are downstream of their TATA-boxes and cover the TSSs (Fig. 3d). The DNA templates used in the cell-free transcription assays encompassed a region spanning −100 to +500 relative to the TSS (+511 for CRISPR B) plus five GC base pairs at either end.

The levels of endogenous Cbp1 in the cell lysate used in the in vitro transcription reactions are insufficient to saturate the large excess of DNA template added to the reaction, allowing us to manipulate the Cbp1-chromatinization of the templates by adding recombinant Cbp1 (Supplementary Fig. 9) while having the full complement of transcription initiation and elongation factors and RNA polymerase

present in the lysate. All three promoters directed the synthesis of run-off transcripts of the expected size (Fig. 3e), while no run-off transcript was detected in negative control reactions including TATA-box mutations of either internal promoter, confirming that the observed transcripts were promoter-specific (Supplementary Fig. 10). The addition of Cbp1 inhibited transcription of internal sense and antisense promoters at 100 nM Cbp1 with near-total inhibition at 300 nM Cbp1. In contrast, the leader promoter remained largely unaffected (Fig. 3e). Addition of Cren7 did not compound the repressive effect of Cbp1. We conducted a slightly modified experiment where the in vitro transcription templates for the three different promoters were combined in a single reaction to facilitate template competition, which mimics the in vivo setting. This reaction was subsequently split into three for separate affinity-purifications of the respective transcripts (Supplementary Fig. 11). The results were consistent consistent with Cbp1 suppressing CRISPR array-internal promoters.

### *cpb1* deletion leads to widespread downregulation in the chromosomal environment of CRISPR arrays

To study the effect of *cpb1* deletion on transcription more widely beyond cryptic promoters, we carried out RNA-seq using a library preparation strategy designed to include also small RNAs such as crRNAs. The RNA-seq data correlated well between samples for the two WT E233S replicates and for two independent *cbp1* strains (Spearman's r = 0.99 and 1.00, respectively, Supplementary Fig. 12). RNA-seq coverage over the CRISPR arrays was dominated by mature crRNAs with the 5' ends of fragments matching the previously described cleavage site for Cas6 endonuclease that processes the precursor transcript into crRNAs[26] whereas the 3'-ends showed signs of exonucleolytic trimming (Supplementary Fig. 13). Notably, crRNA abundance appeared to strongly decline in the 3' half of the CRISPR arrays (Fig. 3c) indicating that premature transcription termination might work as a mechanism to enrich crRNAs derived from more recently integrated spacers at the 5' end of CRISPR arrays in species where CRISPR arrays are extensively long[11]. This effect appears to be Cbp1-independent as the *cbp1* deletion strains showed the same decline in crRNA abundance towards the 3' half of the CRISPR arrays (Fig. 3c).

To assess the wider effects of *cbp1* deletion on transcription, we carried out differential expression analysis for 2655 coding genes with detected expression plus the two CRISPR arrays. 33 and 39 genes were significantly up- and downregulated, respectively (*P*adj < 0.01). CRISPR2 was significantly downregulated (*P*adj < 0.006) and potentially also CRISPR1 (*P*adj < 0.11), in aggrement with the Cappable-seq data. We noted an uneven genome-wide distribution of significantly downregulated genes with strong clustering around the CRISPR arrays (Fig. 3f). Because of this consistent downregulation of transcription in a larger region, we wondered whether this effect could be connected to the chromosome architecture in *S. islandicus* REY15A. Takemata and Bell recently mapped chromosomal interaction domains (CIDs) in *S. islandicus* REY15A using chromosome conformation capture experiments with cells grown under similar growth conditions as in our experiments (exponential growth phase, media supplemented with sucrose and peptide source)[27]. The CRISPR arrays are located within one ~64 kb CID. Significantly downregulated genes were highly enriched within this CID (31 out of 64 genes). Adjusted for predicted operon structures, 18 out 39 predicted operons within this CID contained at least one downregulated gene (Fisher test, adjusted *p*-value < 1*10⁻²⁶, Bonferroni multiple testing correction for all CIDs, up- and down-regulation). This CID-wide downregulation of transcription was also visible in the Cappable-seq data with the exception of the CRISPR array internal TSSs that were upregulated as mentioned above (Supplementary Fig. 14). Overall, RNA-seq and Cappable-seq corresponded well with 14 genes shared between the

28 and 32 genes (Their primary TSSs in the case of Cappable-seq) significantly differentially regulated, respectively, out of 892 genes shared between both data sets (Fisher's exact test *p* < 1*10⁻¹⁴).

### Cbp1 enhances CRISPR array transcription in a minimal system

The experiments with the *cbp1* deletion strain suggest that Cbp1 stimulates CRISPR array expression. To test whether this effect is independent of the CRISRP leader promoters, we cloned chimeric transcription templates by fusing the well characterised SSV1 T6 model promoter to the first 511 bp of *S. solfataricus* CRISPR A and B arrays containing eight repeat sequences. To test a potential orientation bias of Cbp1, we compared templates with the CRISPR B fragment cloned in either orientation. EMSA experiments validated Cbp1 binding to these templates with a saturation of Cbp1 binding occurring at 300 nM (Supplementary Fig. 15). To test Cbp1 stimulation under rigorously defined conditions, we carried out in vitro transcription assays in a minimal, reconstituted assay with recombinant transcription initiation factors. Here we ensure single-round transcription by synchronising initially transcribing complexes at register +6 by omitting CTP and UTP. RNA synthesis commences by the addition of CTP and UTP in the presence of a large excess of a TFB variant that inhibits RNAP reinitiation[22,28,29]. In vitro transcription using the CRISPR B templates resulted in the synthesis of 516nt run-off transcripts within 2 min with good processivity, i. e. only weak partial transcript patterns indicative of elongation pausing (Fig. 4b). The addition of Cbp1 (300 nM) stimulated transcription. This stimulation could be observed using both templates, i. e. independently of the orientation of the CRISPR array and Cbp1-binding sites. The transcript profiles suggest that Cbp1 enhances the overall processivity of transcription. To rule out that direct contacts between the RNAP in the pre-initiation complex and the promoter-proximal Cbp1 increases transcription at the level of initiation, we tested a template with the first repeat sequence randomised to block Cbp1 binding (Δrepeat1). Cbp1 stimulated transcription to a similar extent from both wild type and Δrepeat1 templates, arguing against a role of Cbp1 for transcription initiation. Crucially, a control transcription template encompassing a T6 promoter fusion to a ~500 bp region lacking Cbp1 binding sites did not show stimulation by Cbp1 (Fig. 4d). Our data show that Cbp1 chromatinization enhances transcription processivity in a minimal transcription system in the absence of elongation factors.

### Cbp1 remains bound to CRISPR arrays after CRISPR activation in response to viruses

The expression of CRISPR arrays is induced by viral infection. This effect has been characterised by transcriptome analyses of *S. islandicus* LAL14/1 infected with the lytic virus SIRV2[16]. We used ChIP-seq of Cbp1 and RNA polymerase to specifically test whether virus infection-mediated activation of CRISPR array transcription is accompanied by changes in Cbp1 binding, by comparing uninfected with SIRV2 infected cells. Control ChIP experiments for RNAP showed increased RNAP occupancy on a type I-A Cas operon in SIRV2-infected cells consistent with previously published RNA-seq data showing up regulation of CRISPR systems[16] (Supplementary Fig. 16). ChIP samples from SIRV-infected cells showed consistently an increased background signal originating from the cell lysate input that somewhat skewed the quantification of Cbp1 occupancy. Despite this, Cbp1 occupancy at non-canonical binding sites appeared to be well correlated between SIRV2-infected and infected cells (Fig. 5). When compared to these non-canonical binding sites, Cbp1 occupancy at CRISPR arrays 1 and 2 appeared be unaffected by SIRV2 infection. While we cannot rule out a global reduction in Cbp1 binding upon SIRV2 infection, our data suggest that Cbp1 remains bound to CRISPR arrays when transcription is activated.

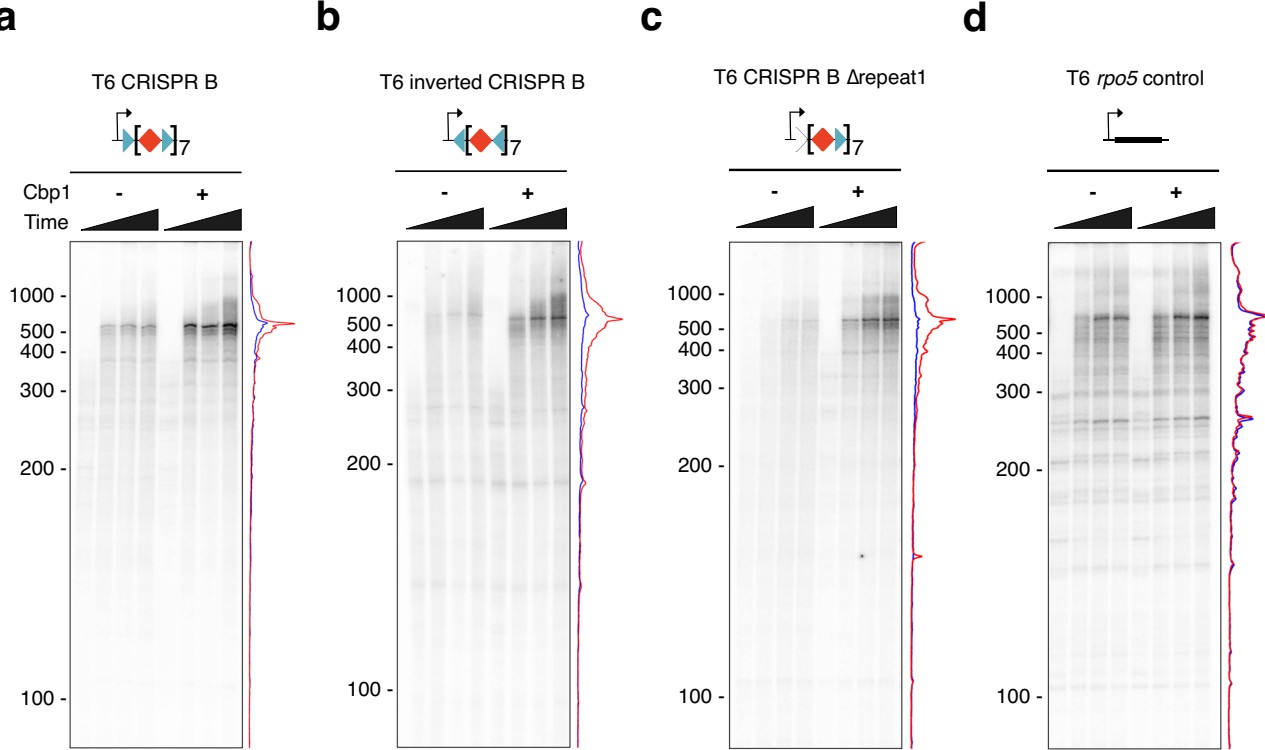

**Fig. 4 | Cbp1 enhances CRISPR array transcription from leader promoters in a minimal transcription system.** Synchronised reconstituted in vitro transcription experiments with the strong T6 promoter fused to the first ~500 bp of *S. solfataricus* P2 CRISPR array B (**a**), the same CRISPR array sequence inverted (**b**), and the same sequence with the first CRISPR repeat randomised (**c**) in the presence or absence of 300 nM Cbp1. As control for a transcription template lacking Cbp1 binding sites we used a fusion of the T6 promoter to the *rpo5* gene from the archaeon *Methanocaldococcus jannaschii*[68] (**d**). Time points 1, 2, 3, and 4 min after release of RNAP from the promoter are shown. Lane profiles for time point 3 min with Cbp1 (red) or without (blue) are depicted on the right of each gel. Representative gels from three replicate experiments are shown.

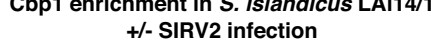
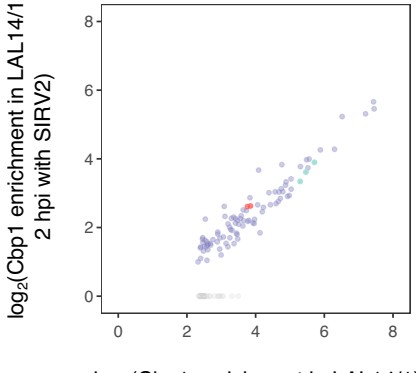

**Fig. 5 | Cbp1 remains bound to CRISPR arrays after CRISPR activation in response to viruses.** Scatter plot showing enrichment calculated for Cbp1 peaks determined for uninfected *S. islandicus* LAL14/1 versus two hours after infection with SIRV2. Average enrichment values for two biological replicates are shown. We obtained consistently lower enrichment values for Cbp1 in SIRV2-infected cells that we attributed to an increased background signal in the IPs originating from the cell lysate input. Broad peaks covering CRISPR arrays 1 and 2 and three non-canonical Cbp1 binding sites in the promoters of CRISPR 3, 4 and 5 (see Supplementary Fig. 14) are highlighted.

## Discussion
### Cbp1 functions as a facilitator of CRISPR array expression
Our results show that the coordinated action of two DNA-binding proteins, the CRISPR array-specific Cbp1 and the general chromatin protein Cren7 facilitate CRISPR array function by specifically enhancing array transcription from the cognate leader promoters and suppressing transcription from cryptic promoters incorporated in CRISPR arrays (Fig. 6). Thereby Cbp1 and Cren7 are preventing interference with CRISPR array expression, e.g. by antisense transcription, and overall safeguard the composition of the crRNA pool. Considering

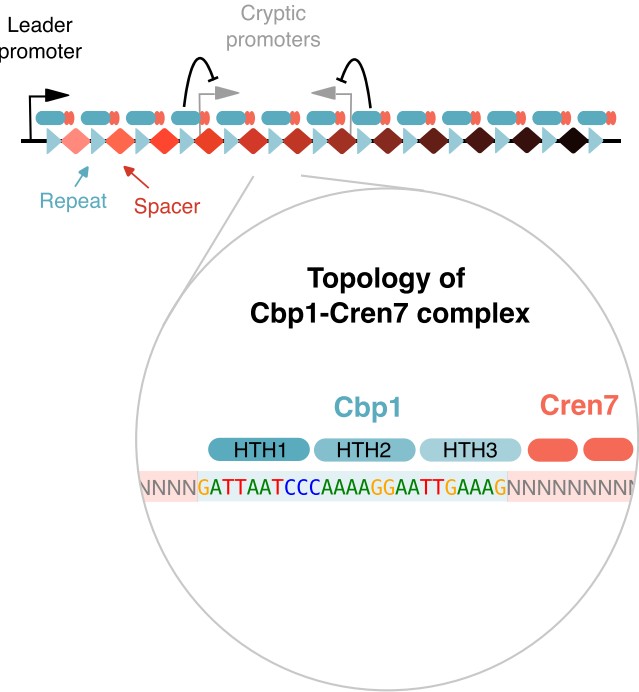

## Organisation of CRISPR array transcription

## Topology of Cbp1-Cren7 complex

**Fig. 6 | Model of Cbp1-Cren7 chromatinization and its role in CRISPR array transcription.** Our data revealed the topology of the Cbp1-Cren7 complex. The three helix-turn-helix motifs of Cbp1 (HTH1 to 3) bind along the CRISPR repeat in 5'->3' direction with HTH3 recruiting multiple Cren7 to the 3'-end of the repeat. Cbp1-Cren7 chromatinization alters CRISPR array transcription by blocking access of the transcription machinery to cryptic internal promoters while facilitating transcription from the leader promoter.

that the length of the leader-containing CRISPR arrays in *S. solfataricus* P2 (arrays A-E) appears to increase with enhanced Cbp1 and Cren7 binding, it is tempting to speculate that Cbp1 chromatinization allowed the emergence of the long CRISPR arrays characteristic for members of the order Sulfolobales genus that have Cbp1. Beyond a function of Cbp1 in array transcription, it may also protect the integrity of the genome, a function generally attributed to chromatin proteins, or more specifically improve the genetic stability of CRISPR arrays by suppressing recombination, which is a high risk due to the highly repetitive nature of CRISPR arrays. For example, Cbp1 binding to each repeat may inhibit homology search during homologous recombination events that could lead to the loss of spacers, and immunity. A recent study suggested that frequent recombination events in CRISPR arrays do occur, including in *S. solfataricus* P2[30].

Cbp1 (and probably Cren7) binding to DNA does not interfere with all processes that utilise the DNA as a template. For instance, Cbp1 does not prevent spacer adaptation by the Cas1-Cas2 machinery in vitro[31]. Our occupancy mapping demonstrates that Cbp1 remains bound to CRISPR arrays during activation of CRISPR systems by SIRV2 infection in LAL14/1 (Fig. 5), suggesting that the Cbp1-Cren7 chromatin is a constitutive part of the *Sulfolobales* CRISPR systems. But most importantly, Cbp1 binding enhances transcription of CRISPR arrays in vivo (Fig. 3 and[9]) and in a minimal system in vitro (Fig. 4), which may sound counterintuitive but shows that Cbp1-Cren7 does not form roadblocks for RNAP during transcription elongation. Rather, Cbp1 is a positive factor that enables CRISPR expression, and it remains to be seen how this is integrated with other types of regulation. One previously identified regulator of CRISPR expression is the cyclic

oligoadenylate-dependent transcription activator Csa3a. Csa3a is thought to control transcription directed by the leader promoters of CRISPR arrays C, D, and E in *S. solfataricus* P2 but not CRISPR arrays A and B, the two arrays with the highest Cbp1 chromatinization that lack the Csa3a binding site within the leader promoter[32].

### Abundant non-canonical Cbp1 binding sites in transposons
Our binding analyses identified many non-canonical, i. e. non-CRISPR, binding sites of Cbp1 in all three Sulfolobales strains (P2, REY15A, LAL14/1), including several strong binding sites in *S. solfataricus* P2 associated with intact or partial IS*C1229* transposons (Figs. 1 and 7) that are not accompanied by Cren7 binding in vivo. Two distinct types of binding sites were present, one within the ORF of the transposase gene (n = 2), and another one within the 3'-flank of the transposon (n = 7). Recent ChIP-seq mapping of transcription regulator PhoB in *E. coli* binding discovered multiple additional binding sites without any apparent functional role[33] and it is likely that this holds true for some of the non-canonical Cbp1 binding sites as well. However, it is unlikely that this is the case for the binding sites within the IS*C1229* transposons given that some of these binding sites show Cbp1 occupancy levels higher than CRISPR repeats with the exception of CRISPR A/B type repeats that are present in *S. solfataricus* P2 but not in *S. islandicus* strains[34]. IS*C1229* belongs to the IS*110* family of insertion sequences, which is characterised by the DEDD family transposase and a circular transposition intermediate formed through homologous recombination between the left and right flanks of the transposon. A curious feature of the IS*110* family transposons is that the strong promoter for transposase expression is formed only upon transposon circularisation[35,36]. If this mechanism is conserved in IS*C1229* transposons, Cbp1 binding in the 3'-flank of IS*C1229* could regulate either access of RNAP to the promoter (and hence transposase expression) or excision of the transposon. Notably, most of the IS*110* family elements in *S. solfataricus* and *S. islandicus* genomes are inactivated, decaying remnants, suggesting that proliferation of IS*110* is kept in check. In mammals, heterochromatin silences repetitive DNA elements and transposons, the role of Cbp1 binding in archaeal transposon function, including transposition remains to be investigated.

### Evolution of chimeric Cbp1-Cren7 chromatinization
The non-canonical Cbp1 binding sites provided insights into the sequence dependence of Cbp1 binding. In particular, they allowed us to identify a core motif of 6 bp which matches the 3'-end of CRISPR repeats (Fig. 1b). This sequence is also conserved in CRISPR repeats found in Desulforococcales species which encode a Cbp1-related protein, Cbp2, that only consists of two HTH motifs[37]. Thus, the 6 bp motif likely represents the core binding site of both Cbp1 and Cbp2.

Cbp1 (and possibly Cbp2) coevolved with Cren7, a general chromatin protein with wider phylogenetic distribution and found in all Crenarchaeota. Our data show that (i) Cbp1 recruits Cren7 in vivo, (ii) the Cbp1-Cren7-DNA complex features at least two Cren7 proteins recruited to the 3' region of CRISPR repeats in vitro, and (iii) the third HTH domain of Cbp1 facilitates the Cren7 interaction (Fig. 6). Multiple Cren7 could directly interact with the Cbp1-DNA complex or, alternatively, Cren7-Cren7 interactions could facilitate the binding of the second Cren7 molecule[38]. A cluster of multiple Cbp1 binding sites might be required to facilitate strong Cren7 recruitment in vivo, as the isolated, strong non-canonical Cbp1 binding sites are not associated with Cren7 enrichment in the genome, but the underlying differences in Cbp1 and Cren7 binding to CRISPR repeats and non-canonical binding sites remain to be solved. Heteromeric chromatin complexes are commonly formed by paralogous proteins such as in the octameric nucleosome complex in eukaryotes or the Alba-Alba2 complex in Crenarchaeota[39]. A heteromeric complex between two unrelated

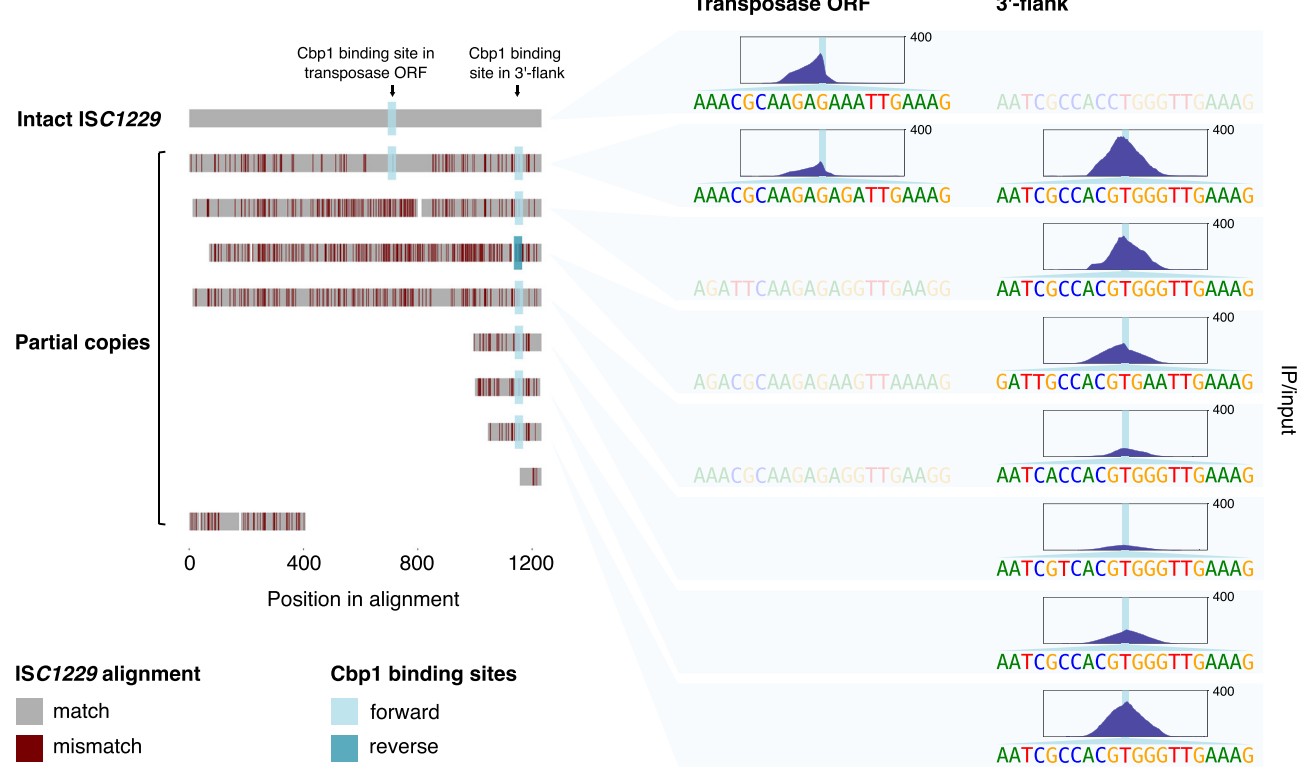

**Fig. 7 | Cbp1 binds to IS110 family transposons.** Alignment of IS*110* family transposons IS*C1229* in the *S. solfataricus* P2 genome. Partial IS*C1229* copies were aligned to the single full-length reference sequence with mismatches highlighted in dark red, insertions (present only in single IS*C1229* copies) not shown. Cbp1 binding sites are highlighted by blue squares according to their orientation (defined by the orientation of Cbp1 binding on CRISPR arrays). The Cbp1 ChIP-seq occupancy at these binding sites is shown on the right (mean signal of two biological replicates) for a 500 bp window centred around the 21 bp Cbp1 binding motif identified at each site. The sequence of the binding sites is shown below each ChIP-seq peak. The corresponding sequences in IS*C1229* copies that do not show Cbp1 binding are depicted as semi-transparent. Coordinates for intact and partial IS*C1229* copies were retrieved from ref. [65]. Coordinates in the *S. solfataricus* P2 genome from top to bottom: 1,745,648–1,746,880 (intact copy serving as reference), 1,449,398–1,450,676, 1,492,743–1,493,952, 2,818,980–2,820,126, 1,761,872–1,763,092, 1,494,821–1,495,060, 1,486,154–1,486,387, 2,832,377–2,832,564, 1,761,726–1,761,802, and 1,476,489–1,476,889. ChIP-seq peaks represent the mean of two biological replicates.

chromatin proteins in *Sulfolobales* that possess a large repertoire of chromatin proteins[2] points to a greater complexity of DNA chromatinization in archaea.

## The fluid boundaries between chromatin and transcription regulators

Several proteins cross the boundary between chromatin and transcription regulators, including H-NS in bacteria, TrmBL2 in *Thermococcus*, and Sa-Lrp and Lrs14 in Sulfolobales[2,40]. Their regulatory mechanisms generally involve nucleation at high-affinity binding sites that is followed by spreading of the proteins along the DNA. While Cbp1 shows some unspecific DNA-binding activity in vitro (see for example Supplementary Fig. 15), our ChIP-seq data show that Cbp1 binding in vivo is limited to CRISPR repeats and other non-canonical Cbp1 binding sites. This remains true even for high affinity binding sites such as the IS*C1229* transposons and the CRISPR array 4 leader in *S. islandicus* LAL14/1. In this respect, chromatinization of CRISPR arrays is mediated by distinct high affinity sites similar to transcription regulators binding to operators but with the unique feature of multiple regularly spaced binding sites occurring in CRISPR arrays. The subsequent, or concomitant, recruitment of Cren7 is comparable to the spreading mechanism, expanding the chromatin structures beyond the CRISPR repeat.

The functioning of chromatin proteins, especially in eukaryotes, is commonly regulated by post-translational modifications, such as methylation and acetylation. In *S. islandicus*, chromatin proteins Alba1, Alba2, Cren7, Sul7d1 and Sul7d2 were found to be methylated[41], with the latter three proteins showing differential methylation patterns depending on the growth phase. Methylation on Lys50 has also been detected in Cbp1[41]. It will be interesting to see whether post-translational modifications on Cbp1 and Cren7 affect their interaction with each other or with DNA.

In future, exploration of Cbp1-Cren7 chromatin in Sulfolobales will benefit from structural analyses of Cbp1-Cren7-DNA CRISPR arrays that will shed light on the topology of transcription unit-specific chromatin in archaea.

## Methods

### Recombinant protein purification

The *S. solfataricus* P2 *cbp1* gene (SSO0454) was cloned into pET-21a(+) (Merck) vector and transformed into BL21 Star (DE3) cells (Thermo Fisher). Proteins were expressed in enriched growth medium for 3 hrs at 37 °C after induction with 1 mM IPTG. Cells were resuspended in N200 buffer (25 mM Tris/HCl, pH 8.0, 10 mM MgCl$_2$, 100 μM ZnSO$_4$, 10% glycerol, 200 mM NaCl, 5 mM β-mercaptoethanol) and disrupted by sonication. Cell debris was removed by centrifugation. The lysate was incubated at 65 °C for 10 min and centrifuged again. The heat-stable supernatant was loaded on a HiTrap Heparin column (GE Lifesciences) and eluted using gradient to 1 M NaCl. The peak fractions were combined, concentrated, and loaded onto a Superose 12 10/300 size exclusion column (GE Lifesciences) equilibrated in N200.

The *S. solfataricus cren7* gene (SSO6901) was cloned into pRSF-1b (Merck) and transformed into Rosetta2 (DE3) pLysS (Merck). 2 h after induction of expression with 1 mM IPTG, cells were harvested and washed in 0.9% (w/v) NaCl. The protein was purified as described

previously using a HiTrap cation exchange chromatography, heat incubation at 70 °C for 30 min, heparin affinity and size exclusion chromatography in the final buffer (50 mM Tris/HCl (pH 8.0), 200 mM NaCl, 10 mM Na-EDTA, 10% glycerol, 10 mM β-mercaptoethanol)[42].

RNA polymerase and recombinant TBP, TFB, and the TFB core-domain variant were produced as described previously[22,43]. All protein concentrations were determined using the Qubit assay system (Thermo Fisher).

### Antisera and antibodies

Polyclonal rabbit antisera against recombinant *S. solfataricus* Cbp1 were produced at Davids Biotechnologie (Germany). Polyclonal rabbit antisera against *S. solfataricus* RNA polymerase (recombinant Rpo4/7 subcomplex), transcription factor TFB and sheep antiserum against Alba have been described previously[22,43,44]. The Alba antiserum was a kind gift of Malcolm White (University of St Andrews, UK). Protein G affinity-purified rabbit antibodies against recombinant *S. islandicus* Cren7 were purchased from CUSABIO (CSB-PA502491LA01FBP, Lot 00911A).

### Strains and *cbp1* gene deletion and cell growth

All genetic experiments were carried out in the *S. islandicus* REY15A strain E233S carrying deletions in the *pyrEF* genes conferring uracil auxotrophy as well as in the *lacS* gene[45]. ChIP-seq experiments were performed with a *cbp1* deletion strain described previously (strain 1)[45]. Since we were unable to revive the original *cbp1* deletion strain from glycerol stocks after prolonged storage time, we reconstructed *cbp1* deletion strains by a CRISPR-based strategy as previously described[46]. The protospacer adjacent motif (PAM) 5′-TCC-3′ (positioned at +150 referring to the start codon of *cbp1*) and the 40-nt sequence immediately downstream of the PAM as protospacer were selected as targeting site. Spacer fragments were generated by annealing of the two complementary oligonucleotides KOcbp-SpF/Kocbp-SpR and inserted into the artificial CRISPR array of pGE1 vector at the *Sap*I sites[47], yielding the interference plasmid pAC-*cbp1*. The donor DNA carrying mutant allele was generated by splicing with overlap extension PCR using the primers Kocbp-Lf/Kocbp-Lr and Kocbp-Rf/Kocbp-Rr. The resulting PCR products were inserted into the *Sph*I and *Xho*I restriction sites of pAC-*cbp1*, yielding the genome-editing plasmid pGE1-*cbp1*. The genome editing plasmid was then transformed into *S. islandicus* E233S competent cells by electroporation and transformants on the plates were validated by colony PCR with primers Kocbp-checkF/Kocbp-checkR. Curing of plasmid pGE1-*cbp1* was achieved by 5-Fluoroorotic acid counterselection. Two independent *S. islandicus* Δ*cbp1* strains (strains 2 and 3) were generated and verified by Sanger DNA sequencing. Strain 2 and 3 were used for TSS-RNA capable-seq experiments (see below).

All *S. islandicus* REY15A strains were grown in Brock medium[48] supplemented with 0.2% sucrose, 0.2% tryptone, and 20 μg/ml uracil at 75 °C in Erlenmeyer shake flasks at 150 rpm. *S. solfataricus* P2 was grown in Brock medium supplemented with 0.2% glucose, 0.1% tryptone likewise.

### SIRV2 propagation

An exponentially growing culture of *Saccharolobus islandicus* LAL14/1[49] was infected with a preparation of *Saccharolobus islandicus* rod-shaped virus 2 (SIRV2). The infected culture was incubated at 76 °C under agitation for 2 days. After the removal of cells (7000 rpm, 20 min; Sorvall 1500 rotor), viruses were collected and concentrated by ultracentrifugation (37,000 rpm, 2 h 30, 15 °C; Beckman Type 45 Ti fixed-angle rotor). The virus titer was determined using a plaque assay.

### Infection experiment and cross-linking

Four 250 mL cultures of *S. islandicus* LAL14/1 were grown in rich medium[18] at 76 °C under agitation for approximately 12 h. When the optical density (OD) reached 0.2, two of the cultures were infected with SIRV2 using a multiplicity of infection (MOI) of 10, while the other two served as uninfected controls (Supplementary Fig. 17). After 2 h post-infection (hpi), 200 mL aliquots were rapidly transferred to flasks, placed on heated magnetic stirring plates and cross-linked with stabilised formaldehyde solution to a final concentration of 0.4%. The cross-linking proceeded for exactly 1 min and the reaction was stopped by adding Tris/HCl pH 8.0 to a final concentration of 100 mM. Samples were cooled down on ice for 5 min and centrifuged (7,000 rpm, 20 min, 4 °C, Sorvall 1500 rotor). Cells were resuspended in 10 mL of PBS and pelleted using Eppendorf Centrifuge 5430 R (7000 rpm, 20 min, 4 °C). Pelleted cells were frozen in liquid nitrogen and stored at −80 °C. Cell density and virus titer were measured at different time points to assess the efficiency of the infection.

### ChIP-seq and ChIP-exo experiments

ChIP-seq data for *S. solfataricus* RNA polymerase subunits Rpo4/7 and transcription factors TFB, Spt4/5, and aCPSF1 were obtained from NCBI GEO superseries GSE141290[22]. All other ChIP-seq experiments were carried out using the same, previously described protocol[22,50] with minor modifications as follows. *S. solfataricus* Cbp1, Rpo4/7, and TFB antibodies were purified from antiserum by Protein A agarose. All *S. solfataricus* antibodies and the *S. islandicus* Cren7 antibody showed good cross-reactivity in immunoprecipitation experiments for the other species. For *S. solfataricus* TFB and *S. islandicus* REY15A Cbp1, TFB, and Rpo4/7 ChIP experiments, 2 μg antibody were incubated with 500 μl cell lysate at 20 ng/μl DNA content. For Cren7 ChIP, the amount of antibody was increased to 4 μg. For ChIP experiments in *S. islandicus* LAL14/1, 2 μg Rpo4/7 or Cbp1 antibodies were incubated with 600 μl cell lysate at 12 ng/μl DNA content.

For ChIP-exo experiments, DNA shearing by sonication was altered to yield fragments within the recommended 200–1200 bp range. 1 ml of cell lysate was incubated with 8 μg Cbp1 or Cren7 antibody overnight at 4 °C and antibodies were captured by further incubation with 50 μl Protein G-beads for 1 h. Beads were washed six times in 1 ml RIPA buffer (50 mM HEPES/NaOH pH 7.6, 1 mM Na-EDTA, 0.7% Na-Deoxycholate, 1% NP-40, 0.5 M LiCl) before two wash steps in 10 mM Tris/HCl pH 8.0. Library preparation was carried out following the ChIP-exo 5.0 method[51] with both the ExA1 and ExA2 adaptor sequences modified by introducing barcodes for dual indexing (ExA1_i5X_58: 5′-AATGATACGGCGACCACCGAGATCTACACN$_8$ACACTC TTTCCCTACACGACGCTCTTCCGATCT-3′ and ExA2_i7X: 5′-[Phos]-CAAGCAGAAGACGGCATACGAGATN$_8$GTGACTGGAGTTCAGACGTGT-GCTCTTCCGATCT-3′, where X and N denote the barcode number and sequence, respectively). Two biological replicates were used for all ChIP-seq and ChIP-exo experiments and deep-sequencing was carried out using Illumina HiSeq 125 cycle Paired-End Sequencing v4 or Hi-Seq 4000 75 Paired-End Sequencing.

### ChIP-seq data analysis

Mapping and normalisation of ChIP-seq data was conducted as previously described[22]. In brief, ChIP-seq paired-end reads were mapped using Bowtie v1.1.2[52] with parameters -v 2 -m 1 -fr over the first 50 nt allowing only for uniquely mapped read pairs to be included. The mapped fragments were sampled to match a normal distribution with a mean of 120 and standard deviation of 18 or in the case of LAL14/1 ChIP-seq data a mean of 150, standard deviation of 20. Normalisation to input was carried out using deepTools bigwigCompare using signal extraction scaling (10,000 bins, 200 bp bin width)[53].

### Cbp1 ChIP-seq peak calling

ChIP-seq peak calling was performed identically for each of the four different strains (*S. solfataricus* P2, *S. islandicus* REY15A derived strain E233S, *S. islandicus* LAL14/1, and *S. islandicus* LAL14/1 + SIRV2) The paired-end alignment files for Cbp1 ChIP-seq data (see above) were

converted to bed file format. Calling of narrow peaks on ChIP-Seq data was performed for each replicate using MACS2 version 2.2.6[54] in the BEDPE mode with a cut-off of q = 0.01 and the call-summits subfunction activated. Input data were used as control sample. MACS2 estimated duplicate reads to be within the expected level. Peaks with summits located within the CRISPR arrays and rRNA loci were removed for subsequent analysis. Matching of peaks between the two replicates was performed based on the peak summit positions differing by maximal 40 bp using BEDTools window[55]. For S. solfataricus P2 Cbp1, 199 matched peaks were further filtered for consistent ranking based on the p-value using the IDR method[56] with an estimated correlation coefficient rho = 0.99 for the reproducible component (global IDR < 0.01) representing a fraction of 0.80 of all peaks after 100 iterations. This resulted in a set of 120 peaks with a global IDR < 0.01. 92 peaks with an average enrichment of >5 (based on MACS2 output) were used for all further data analysis (Supplementary Data 1).

For S. islandicus E233S Cbp1, IDR filtering resulted in an estimated correlation coefficient rho = 0.99 for the reproducible component (global IDR < 0.01) representing a fraction of 0.71 of all peaks after 100 iterations. This resulted in a set of 352 peaks with a global IDR < 0.01. 157 peaks with an average enrichment of >5 were used for all further data analysis (Supplementary Data 2).

For S. islandicus LAL14/1 Cbp1, IDR filtering resulted in an estimated correlation coefficient rho = 0.94 for the reproducible component (global IDR < 0.01) representing a fraction of 0.68 of all peaks after 100 iterations. This resulted in a set of 150 peaks with a global IDR < 0.01. 118 peaks with an average enrichment of >5 were used for all further data analysis (Supplementary Data 3).

For the SIRV2 infected LAL14/1 cells a lower enrichment threshold had to be applied because of the increased genomic DNA background in the ChIP samples. IDR filtering resulted in an estimated correlation coefficient rho = 0.90 for the reproducible component (global IDR < 0.01) representing a fraction of 0.45 of all peaks after 100 iterations. This resulted in a set of 68 peaks with a global IDR < 0.01, all with an average enrichment of >3 used solely for quality control (Supplementary Data 4).

Supplementary Fig. 18 shows scatter plots depicting the peaks called for individual replicates and their correlation. Replicates for all three species and conditions (with or without SIRV2) showed an $R^2$ value of 0.99 for the matched peaks as well as the IDR-filtered subsets of the matched peaks.

To quantify the broad region of enrichment of Cbp1 at CRISPR arrays 1 and 2 in LAL14/1, we used MACS2 version 2.2.6[54] in the BEDPE mode with a cut-off of q = 0.01 and the --broad subfunction activated and --min-length 6000.

To compare Cbp1 peak enrichment values between uninfected and SIRV2-infected (2 hpi) S. islandicus LAL14/1 cells, we matched the set of 118 IDR- and minimum enrichment-filtered peaks from uninfected cells with a set of 155 peaks with reproducible summit positions (within 40 bp using BEDTools window) for the two SIRV2-infected cell replicates without additional IDR-filtering. Matching the datasets using BEDTools window with 40 bp maximal average summit distance resulted in 92 common peaks (excluding the two broad peaks across CRISPR arrays 1 and 2) between the two growth conditions.

## DNA sequence motif identification
To identify DNA sequence motifs within the non-canonical binding sites of Cbp1, DNA sequences covering genomic intervals covering 80 bp (S. solfataricus P2) or 50 bp (S. islandicus REY15A) on either side of Cbp1 ChIP-seq peak summits (for peaks with minimum 5-fold enrichment) were retrieved using BEDTools getfasta[55]. Motif search was conducted using the MEME software version 4.11.2 with one occurrence of the motif per sequence[46]. A 0-order background model based on the respective genome sequences was employed.

## Differential binding analysis of S. islandicus REY15A Cren7
In order test for differential binding of Cren7 in S. islandicus REY15A E233S and the cbp1 deletion strain1 ChIP-seq data, the CSAW package v1.26.0[57] in R v.4.1.1[58] was used for consecutive 30 bp windows and 250 bp maximal fragment length with a minimal count of 20 reads across the four libraries with two biological replicates per strain. A Benjamini–Hochberg multiple testing correction was applied for the differential binding results of the 30 bp windows. Windows overlapping with CRISPR arrays and non-canonical Cbp1 binding sites(peak summits from MACS2 peak calling) were obtained using BEDTools window[55] with maximal 30 bp distance (bedtool window -c -w 30).

## ChIP-exo data analysis
ChIP-exo reads were mapped using Bowtie v1.1.2[52] with parameters -v 2 -m 1 –fr allowing only for uniquely mapped read pairs to be included. Bam files with alignments were filtered for first read in proper pairs (samtools 1.10 with view -f 66 -b[59]) and the read 5′-end coverage per strand was calculated using BEDTools genomecov[55]. The geometric mean of 5′-end coverage for two biological replicates was calculated and scaled to 1x genome coverage. Aggregate plots were calculated for 198 CRISPR repeat in CRISPR arrays A and B and the 92 non-canonical binding sites associated with Cbp1 ChIP-seq peaks.

## Operon prediction for S. islandicus REY15A
Operon structures for S. islandicus REY15A were predicted using operon-mapper[60] with genome sequence and annotation obtained from NCBI (ASM18955v1).

## Cappable-seq
Cappable-seq data for S. solfataricus P2 were described in Blombach et al.[22] (NCBI GEO superseries GSE141290) and data for S. islandicus REY15A E233S (two biological replicates) and Δcbp1 strains 2 and 3 were generated likewise. In brief, cells were grown to exponential phase, cultures were mixed with 2 volumes of pre-cooled RNAprotect Bacteria Reagent (Qiagen), and cells were collected by centrifugation (5 min at 4000 × g at 4 °C). Small RNA preparations (20–200 nt length) were carried out using the mirVana miRNA isolation kit (Ambion/Thermo Fisher) following the manufacturer's protocol. The library preparation including an enrichment step for 5′-triphosphorylated RNAs by capping the RNAs with 3′-desthiobiotin-TEG-GTP (NEB)[23] and subsequent deep sequencing on a Illumina NextSeq 500 system with 75 bp read length were conducted at Vertis Biotechnologie (Germany) as previously described[22].

## Cappable-seq data analysis
Data processing of S. islandicus REY15A Cappable-seq data was carried out as described previously[22]. In brief, poly(A)-tails and 3′-adaptors in the reads were removed, reads were aligned to the S. islandicus REY15A genome and the resulting bam files were merged and 5′-end coverage was calculated using BEDTools genomecov[55]. Bedgraph output was converted into bigwig file format.

For the identification of TSSs we proceeded as follows. We considered positions with an absolute increase of >20 read 5′-ends and a relative increase of >5-fold compared to the preceding position as candidate TSSs for each individual Cappable-seq sample and strand. Candidate TSSs reproducible across at least two samples (within or across conditions) were called. TSSs within the cbp1 gene and its promoter were removed. To take into account more dispersed transcription initiation over a short region, we clustered called TSSs occurring within less than 5 bp distance and took the centre position of these TSS clusters as final TSS position. For the differential expression analysis, the Cappable-seq signal was integrated over a 11 bp window around the centre position of each TSS cluster and differentially expressed TSSs were called by Deseq2 v1.40.2[61] with a Padj 0.01 cut-off.

## RNA-seq *S. islandicus* REY15A

To gain insight into the differential expression of both coding genes and CRISPR arrays in Δ*cbp1* strains versus the parental strain E233S, we followed a similar strategy as previously described by Quax et al.[16]. *S. islandicus* REY15A E233S (two biological replicates) and Δ*cbp1* strains 2 and 3 were grown to exponential phase, cultures were mixed with 2 volumes of pre-cooled RNAprotect Bacteria Reagent (Qiagen), and cells were collected by centrifugation (5 min at 4000 × g at 4 °C). Total RNA preparations were carried out using the mirVana miRNA isolation kit (Ambion/Thermo Fisher) and genomic DNA was removed using the TURBO DNA-free kit (Ambion/Thermo Fisher) following the manufacturer's protocol. Ribosomal RNA was removed using the Pan-Archaea RNA-seq riboPool kit (siTOOLS biotech) and the RNA was subsequentially fragmented using Ambion RNA Fragmentation Reagents (Thermo Fisher) according to manufacturer's protocol. Fragments in the size range of 17-200 nt were subsequently purified using RNA Clean & Concentrator-5 kit (Zymo Research). Purified RNA fragments were treated to remove 3´ end blocking groups (in particular 2′,3′-cyclic phosphate from crRNAs) using 10 U T4 polynucleotide kinase (NEB) for 1 h 30 min at 37 °C in 70 mM Tris/HCl pH 6.5, 10 mM MgCl$_2$, 1 mM DTT, 20 U SUPERaseIn RNase Inhibitor (Ambion) in 20 µl reactions. To phosphorylate the 5´-ends, additional 10 U polynucleotide kinase (NEB) in presence of 1 mM ATP, 10 U SUPERaseIn RNase Inhibitor (Ambion), 1x T4 PNK buffer (NEB) were added to the reactions in a total reaction volume of 50 µl and the reactions were incubated for 40 min at 37 °C. Finally, the treated RNA fragments were ethanol-precipitated, resuspended in 6 µl of H$_2$O and converted to Illumina sequencing libraries using NEBNext Small RNA Library Prep Set for Illumina kit (NEB) with 12 cycles of PCR-amplification. Libraries were analysed on High Sensitivity dsDNA chips (Agilent Technologies), pooled for to equimolar concentration, and the library pool was isolated from a 1x TBE polyacrylamide gel to remove remaining adaptor dimers. The library pool was sequenced on a NovaSeq SP flow cell (Illumina) with paired-end 50 bp reads.

## RNA-seq data analysis

5′- and 3′- end adaptor and primer sequences in the reads were removed using cutadapt 4.1[62] with settings -e 5 -m 25 -a AGATCGG AAGAGCACACGTCTGAACTCCAGTCACNNNNNNATCTCGTATGCCGT CTTCTGCTTG -A GATCGTCGGACTGTAGAACTCTGAACGTGTAGA TCTCGGTGGTCGCCGTATCATT where NNNNNN stands for barcode of the indexing primer. Subsequently, reads were aligned to the *S. islandicus* REY15A genome (bowtie 1.0.0 with settings --allow-contain -y -v 2 -m 1 --strata --best[52]) and the resulting bam files were filtered for proper read pairs using samtools[59]. DNA fragment coverage was calculated using BEDTools v2.30.0 genomecov[55] scaled to count per million fragments (read pairs). Bedgraph output was converted into bigwig file format. The arithmetic mean for two biological replicates was calculated in R.

For differential expression analysis, fragments overlapping with coding genes or CRISPR arrays were counted using featureCounts v2.0.6[63] with settings -s 1 -d 25 -p --countReadPairs -B -t repeat_region,CDS -g ID (CRISPR arrays being annotated as "repeat_region"). Count data were imported into R and differentially expressed genes and CRISPR arrays were called by Deseq2 v1.40.2[61] with a *P*adj 0.01 cutoff. Circos plots were generated using the circlize package v0.4.15 in R[64] with CID coordinates obtained from Takemata and Bell (2021)[27]. Heatmaps for crRNA coverage were generated using deepTools v3.5.1[53].

Quantification of transcripts coding for chromatin proteins in *S. islandicus* REY15A was obtained using featureCounts[61] as described above but limited to CDS (excluding the CRISPR arrays). Paired-end RNA-seq data for *S. solfataricus* P2 cells during exponential growth phase were published previously (Blombach et al.[22], NCBI GEO GSE141290) and counted likewise. Raw fragment counts were normalised against gene length and subsequently scaled to 1 million fragments to obtain transcript per million values (TPM) (Supplementary Table 2).

## EMSA for templates with single Cbp1 binding sites

Double-stranded DNA templates bearing a 24 or 25 bp CRISPR repeat plus 20 bp flanking sequence on either side (or as otherwise indicated, see Supplementary Data 5) were assembled by hybridisation of complementary oligonucleotides with one oligonucleotide being radiolabelled with [γ-$^{32}$P]-ATP by T4 polynucleotide kinase (NEB). 15 µl EMSA samples contained 800 pM DNA template, 10 mM MOPS pH 6.5, 100 mM KCl, 10 mM MgCl$_2$, 2 µg bovine serum albumin (NEB), 10% glycerol (v/v) and 5 mM DTT as well as in total 1.5 µl of Cbp1 and Cren7 preparations in N200 buffer. Samples were incubated for 5 min at 75 °C and resolved on 6% native Tris-Glycine gels containing 2.5% glycerol and 1 mM DTT. Gels were dried and the signal was detected on BAS storage phosphor screens scanned on a Typhoon FLA 9500 scanner (GE Lifesciences).

## DNase foot-printing assays

DNase I foot-printing assay conditions were identical to those in EMSA experiments with the following modifications: the dsDNA concentration was raised to 10 nM with Cbp1 concentrations raised to 12.5 or 25 nM and Cren7 concentrations raised to 50 nM as indicated. After incubation for 5 min at 75 °C, 15 µl samples were allowed to cool to room temperature and 1 µl RQ1 DNase (Promega) diluted to 0.1 U/µl in storage buffer (10 mM Hepes pH 7.5, 50% glycerol (v/v), 10 mM CaCl$_2$, 10 mM MgCl$_2$) was added to the samples. Samples were incubated for 10 min at room temperature before the addition of 16 µl formamide sample buffer (95% deionised formamide, 18 mM EDTA, 0.025% SDS). Samples were heated for 5 min at 95 °C before loading onto an 12% polyacrylamide, 7 M Urea, 1× TBE sequencing gel. Gels were dried and the signal was detected on BAS storage phosphor screens scanned on a Typhoon FLA 9500 scanner (GE Lifesciences).

## EMSA for templates with multiple Cbp1 binding sites

To test Cbp1 binding to the 600 bp DNA templates used in reconstituted in vitro transcription assays, we used identical buffer conditions and protein concentrations as in the transcription assays omitting nucleotides, RNA polymerase and transcription factors. Samples were incubated for 5 min at 65 °C and 12 µl were loaded onto a 1.5% agarose gel. Gels were run for 16 hrs at ~0.9 V/cm in 1xTAE buffer and post-stained with ethidium bromide. Gels were visualised on a Typhoon FLA 9500 scanner (GE Lifesciences).

## Cross-linking of Cbp1 and Cren7

Cbp1 and Cren7 aliquots were buffer-exchanged into 25 mM HEPES/NaOH pH 7.5, 200 mM NaCl by ultrafiltration. DNA Templates were annealed in 10 mM HEPES/NaOH pH 7.5, 50 mM NaCl. Cbp1, Cren7, and template DNA concentrations were kept at 2.5 µM. Samples were incubated at 75 °C for 5 min. 1 mM bissulfosuccinimidyl suberate (BS$^3$) cross-linker (ThermoFisher) was added to each sample and incubated at 25 °C for 30 min. Cross-linking reactions were quenched by the addition of 50 mM Tris/HCl (pH 8.0) and samples were resolved by SDS-PAGE followed by Coomassie Blue staining.

## Cell-free transcription assays

DNA templates for cell-free transcription were PCR-amplified as outlined in Supplementary Data 5. *S. solfataricus* cell-free transcription assays were adapted from[22] for multi-round transcription conditions. 30 µl samples contained 20 µl cell lysate in 10 mM MOPS pH 6.5, 10 mM MgCl$_2$, 60 ng DNA template, 2 mM spermidine supplemented with 10 mM rNTPs, trace amounts of [α-$^{32}$P]-UTP and the indicated amounts of Cbp1 and Cren7. Samples were incubated at 70 °C for 4 min before being placed on ice. Transcripts were isolated by affinity purification

using 3′-biotinylated antisense oligonucleotides matching the first 25 nt of the transcripts (based on the mapped TSS, Supplementary Data 5) as previously described[22] and resolved on 8% polyacrylamide, 7 M Urea, 1× TBE sequencing gels. Gels were dried and the signal was detected on BAS storage phosphor screens scanned on a Typhoon FLA 9500 scanner (GE Lifesciences). For the competition assay (Supplementary Fig. 11), the DNA templates were generated using primers with amino C12 modified 5′-ends in order to suppress the generation of non-specific longer transcripts. 90 μl reactions were of identical composition as described above including 60 ng of each of the three DNA templates. Samples were incubated at 75 °C for 4 min before being placed on ice. Reaction were split into three for separate affinity purifications of the three different transcripts with their respective antisense oligonucleotides as above.

### Reconstituted in vitro transcription assays

As template for reconstituted in vitro transcription assays, the initial 511 bp of *S. solfataricus* P2 CRISPR arrays A and B were amplified by PCR from genomic DNA (Supplementary Data 5). The forward primer for the PCR encompassed a 46 nt sequence matching the strong viral model promoter T6 (positions −46 to −1 relative to TSS) fused to a priming region matching the initial 30 nt of the transcribed region of CRISPR array B. To test transcription through an inverted CRISPR array B (antisense direction), a corresponding construct was designed encompassing region −46 to +2 of the T6 promoter fused to the inverted 511 bp region of CRISPR array B. PCR products were cloned into vector pGEM-T (Promega). CRISPR A and B start naturally with a 6 nt cassette consisting of adenines and guanines that we used for synchronisation of transcription (see below). For the inverted CRISPR B construct, we altered the initial 6 nt to the same sequence using site-directed mutagenesis. All sequences were amplified by PCR from the plasmids using primers carrying 5 guanine nucleotides at their 5′-ends. Randomisation of the first repeat sequence in the T6 CRISPR B fusion was achieved by site-directed mutagenesis.

45 μl Transcription reactions containing 90 ng DNA template, 1 μg RNA polymerase, 1 μM TBP, 0.125 μM TFB, and the indicated concentrations of Cbp1 and Cren7 in transcription buffer (10 mM MOPS pH 6.5, 10 mM $MgCl_2$, 105 mM KCl, 10% glycerol (v/v), 0.8 μg BSA, 10 mM DTT, 5 μg/ml heparin, 2 mM spermidine) supplemented with 100 μM ATP/GTP were incubated for 5 min at 65 °C to allow for the formation of initially transcribing complexes. A single round of transcription was initiated by the addition of 250 μM ATP/GTP/CTP, 25 μM UTP supplemented with trace amounts of [α-$^{32}$P]-UTP and 5 μM TFB core-domain variant. 10 μl were withdrawn at the indicated time points and mixed with 200 μl stop solution (0.3 M Na-Acetate pH 5.2, 10 mM Na-EDTA, 0.5% SDS, 150 μg/ml GlycoBlue (ThermoFisher). After extraction with Acid-Phenol-Chloroform (ThermoFisher) and ethanol precipitation, the pellets were dissolved in 10 μl formamide sample buffer (95% deionised formamide, 18 mM EDTA, 0.025% SDS) and heated for 5 min at 95 °C before loading onto an 8% polyacrylamide, 7 M Urea, 1× TBE sequencing gel. Gels were dried and the signal was detected on BAS storage phosphor screens scanned on a Typhoon FLA 9500 scanner (GE Lifesciences).

### Immunodetection of Cren7 in Δcbp1 strains

*S. islandicus* REY15A Δ*cbp1* strains 2 and 3 (2 biological replaces per strain) and the parental strain E233S (4 biological replicates) were grown to mid-exponential growth phase (O.D.$_{600}$ = 0.13 to 0.18) in 50 ml cultures. Cells were harvested by centrifugation and stored at −80 °C. The cell material was resuspended in 500 μl TK150 buffer (20 mM Tris/HCl pH 8.0, 150 mM KCl, 2.5 mM $MgCl_2$, 100 μM $ZnSO_4$) supplemented with 50 μg/ml DNase I and disrupted by sonication. After the removal of cell debris, the protein content of the lysate was estimated by using the Qubit protein assay (Thermo Fisher) and the lysate concentration was adjusted with TK150 buffer to 0.5 mg/ml.

Samples were resolved using SDS-PAGE on a 14% Tris-Tricine gel and blotted onto a nitrocellulose membrane (GE Lifesciences). Cren7 was detected using a 1/1000 dilution of rabbit anti *S. islandicus* cren7 antibody (CUSABIO, Lot O0911A) as primary antibody and donkey anti-rabbit IgG Dylight680 (Bethyl Laboratories) as secondary antibody (1/10,000 dilution). As loading control, we detected the general chromatin protein Alba with sheep *S. solfataricus* Alba antiserum as primary antibody (kindly provided by Malcolm White, University of St Andrews, UK) with donkey anti-sheep IgG Alexa488 (Thermo Fisher) as secondary antibody (1/2000 dilution). Blots were scanned on a Typhoon FLA 9500 scanner (GE Lifesciences) and overlays were produced in ImageQuant TL 1D v8.2.0 (GE Lifesciences).

### Identification and alignment of IS110 family transposases

Transposon data were obtained from[65] and the ISfinder database[66]. Transposon sequences were aligned using MUSCLE[67] with insertions specific to single copies of the intact transposon removed. Fasta alignment files were imported into R and visualised using the msavisr function in the seqvisr package v0.2.7 (https://doi.org/10.5281/zenodo.6583981).

### Reporting summary

Further information on research design is available in the Nature Portfolio Reporting Summary linked to this article.

## Data availability

The sequencing data generated in this study (ChIP-seq, ChIP-exo, RNA-seq and Cappable-seq) and the processed data have been deposited at NCBI GEO under accession code GSE226026. RNA-seq and Cappable-seq data and RNAP, TFB, Spt4/5, aCPSF1 ChIP-seq data for *S. solfataricus* P2 were obtained from NCBI GEO GSE141290[22]. The sequence of ISC1229 was obtained from the ISfinder database[66] [https://isfinder.biotoul.fr/scripts/ficheIS.php?name=ISC1229]. Raw data for plots in Figs. 2c, 3b, and 5 and uncropped gel images are provided in the Source Data file. Source data are provided with this paper.

## Code availability

The analysis code is available at Zenodo [10.5281/zenodo.10557037].

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

## Acknowledgements

We thank Malcolm White (University of St Andrews, UK), Remus Dame (Leiden University, NL), Michael Terns (University of Georgia, GA) and Roger Garrett (University of Copenhagen, DK) for valuable advice on the project and Stephen Bell (Indiana University Bloomington, USA) for sharing the CID coordinates. Research in the RNAP laboratory at UCL is funded by a Wellcome Investigator Award in Science 'Mechanisms and Regulation of RNAP transcription' to FW (WT 207446/Z/17/Z).

## Author contributions

F.B., M.S., J.C., X.F., D.Baq., T.F., D.K.P., D.Bar., M.K., Q.S. and F.W. F.B., M.S., J.C., X.F., D.Baq., T.F., D.K.P., D.Bar. designed and performed experimental work. F.B. performed computational data analyses. M.K., Q.S. and F.W. acquired funding and supervised the work. F.B. and F.W. coordinated the project and wrote the manuscript with input from all authors.

## Competing interests

The authors declare no competing interests.
