## [Peer Review File NEW · Nature Communications]

Cbp1 and Cren7 form chromatin-like structures that ensure efficient transcription of long CRISPR arraysReviewer #1 (Remarks to the Author):

CRISPR arrays form an adaptive immune system that necessitates capture and retention of invader sequences. The capture of random, or near random sequences from invaders invokes concerns of capturing sequences that would drive transcription (or other molecular events) from internal spacers within CRISPR arrays. Mechanisms to ensure leader-driven expression of CRISPR arrays to promote adaptive immunity must be balanced with efforts to limit spurious transcription from within arrays. Evidence that nucleoid-binding proteins (NAPs) specifically associate with CRISPR arrays argues that specific NAPs may play critical biological roles in controlling CRISPR expression.

Blombach et al. report on the recruitment and impact of Cbp1 and Cren7 binding to the CRISPR arrays in multiple species of crenarchaeota (the manuscript switches between several species with minimal discussion of the differences in biology of such). In vivo ChIP profiles (from only duplicate samples) form the basis of the primary analyses which are bolstered by in vitro biochemistry and transcription assays. Binding of Cbp1 recruits Cren7 that collectively promotes transcription from leader while limiting internal promoters from firing within CRISPR arrays. However, in impact of Cren7 binding remains unclear given the strong impacts of Cbp1 alone.

The collective data suggests a few main findings that each have some caveats:

1) Cbp1 recruits Cren7 in a sequential manner to bind CRISPR arrays.

Caveats: The authors consistently use the term chromatinization, suggesting a robust level of DNA binding that might approach saturation or invoke a specific nucleoid structure with regulatory potential. However, no data is provided to validate the use of chromatinization as the term is normally applied. Cbp1 and Cren7 do bind the CRISPR arrays, but neither the in vivo nor in vitro data provided imply the formation of a regular structure that would meet the normal definition of chromatinization. The image in Figure 7 that shows a near complete saturation of the CRISPR array should be supported by experimental evidence. The ensemble average of billions of genomes through ChIP is not sufficient to define a saturated protein environment due to Cbp1 and Cren7. Also, given that some arrays are not bound by Cbp1 suggests that the general use of arrays should be replaced throughout with Cbp1 bound arrays. The central text of pg 15 highlights this fact, but it is not addressed until page 15.

2) Cbp1 and Cren7 binding to CRISPR arrays suppresses cryptic transcription while also promoting leader template transcription.

Caveats: While an impact of Cbp1 binding is conclusive with regards to transcription in vitro, the manuscript lacks any evidence Cren7 binding influences transcription from leader or internal CRISPR promoters in vivo or in vitro.

3) A suggestion is made that viral infection differentially regulates both Cbp1 and RNAP recruitment into the CRISPR arrays to impact expression of the array.

Caveats: a true comparison is not available due to contamination issues. Given the failure to be able to correctly analyze the data, this entire section is speculative and uninformative. Either do the experimentation again and provide a proper statistical analysis or remove completely. As written, the authors are making a major claim with massaged and partial data.

Major concerns require experimental and writing improvements before publication can be endorsed.

Major:

1. The use of multiple species complicates interpretations and is often confusingly written. As a single example, the causal relationship between Cbp1 and Cren7 recruitment into the CRISPR arrays in *S. solfataricus* is inferred using a Cbp1 deletion strain in *S. islandicus*. The authors should minimally include ChIP data of recruitment of both proteins in *S. islandicus* strain. Figure 1d figure title is misleading to use *S. islandicus* WT vs Δ cbp1 when the WT strain is not *S. islandicus* being plotted. Cren7 could potentially be recruited differently in *S. islandicus* compared to *S. solfataricus*. Equating the two different species without very clear clarification is simply not scientifically sound. While this one example is listed, many problems of a similar nature are relevant throughout the manuscript due to the change in species (and changes in making new strains to correct for unexpected deletions).

2. The use of duplicates only for all ChIP analyses complicates interpretations. The authors must provide some evidence regarding the congruence of the peaks called for replicate #1 and #2. An

average of 1 and 99, or 49 and 51, is equal to 50, but the former has a lot of concerns compared to the latter. Inclusion of a simple XY plot of peaks called across the genome in each ChIP experiment for replicate #1 versus replicate #2 would provide more confidence. An R2 value of 0.9 or above would instill confidence in the results. Would the data, and interpretations from such, look nearly identical if only one replicate was used? What, if any, meaningful differences were revealed in the ChIP replicates? Inclusion of a third replicate, with retention and analyses based on using only peaks identified in 2/3 samples would more closely match the norms of publications at this level and vastly improve confidence in the approach presented here.

3. The absence of a traditional RNA-seq analyses is a major disappointment considering the level of interpretation made regarding recruitment of RNAP and transcription factors to CRISPR arrays that is measured by ChIP. Cappable seq is not a substitute for the impacts that would be revealed by RNA-seq.

4. The authors suggest the Δ HTH1 and Δ HTH3 variants do not impact expression and thermostability without providing supporting evidence. While the impacts of such deletions could be interpreted as the authors present, alternative interpretations would be partially eliminated by inclusion of additional data. Figure 2c should be accompanied by a statistical analysis between the means of each condition (WT, Δ HTH1, and Δ HTH3), comparing each category (nt change). An ANOVA Pairwise Comparisons for parametric data or Dunn's test for non-parametric data are likely appropriate.

5. In vitro transcription assays demonstrate qualitative impacts of Cbp1 and Cbp1/Cren7 DNA binding on transcription from different promoters identified by ChIP binding of transcription components. The absence of any quantification is perplexing and would improve a revised version of the manuscript. The selection of internal sense and internal antisense templates from CRISPR F but the leader template from CRISPR B without proper explanation is perplexing. A competitive experiment of mixing templates with different promoters (leader versus internal) within the same reaction is a clear and obvious addition to the manuscript. Addition of Cbp1/Cren7 under a condition of mixed templates is predicted to differentially impact internal promoters. Furthermore, existing data predict only Cbp1 is necessary to prevent transcription initiation from an internal promoter begging the question of whether or how Cren7 binding may impact repression of internal transcription. The lack of a direct role for Cren7 in vitro, and no evidence of transcription levels in vivo, demands a change of title. Figure 4 also only implicates the activity of Cbp1, not Cren7.

6. The authors state that "Recruitment of Cren7 could be mediated by direct physical interaction between Cbp1 and Cren7 or by Cbp1-induced topological changes in the DNA template facilitating Cren7 binding". The authors conclude "Our data thus demonstrate that Cbp1 recruits Cren7 through direct physical interactions". While this may be true, Figure 1f suggests that the presence of DNA is required for Cbp1 and Cren7 interactions. When DNA is not present (first lane after the ladder) or the DNA is non-specific (last lane), the proteins are not interacting. This suggests that DNA is a very important component in these interactions and thus one cannot rule out "Cbp1-induced topological changes in the DNA template facilitating Cren7 binding". Additionally, Figure 2b suggests that DNA is important, as removing Cren7 binding to the DNA abolishes complex formation.

Minor:

The new(er) naming system of the species employed should be used. Minor spelling and grammatical errors persist throughout.

Reviewer #2 (Remarks to the Author):

This manuscript deals with the regulation of CRISPR repeat-spacer array transcription by the CRISPR array binding protein 1 (Cbp1) in Sulfolobales archaea. These arrays are of extraordinary length raising the question how efficient transcription could reach the more distal parts of the arrays.

In this context, two previously made seemingly contrary observations existed: that Cbp1 binds to

the CRISPR repeat sequences where it could interfere with transcription, while experimental data showed the opposite, that Cbp1 overexpression actually increased CRISPR-RNA levels.

Technically, the authors mapped Cbp1 binding globally at single-nucleotide resolution by ChIP-seq combined with 5'→3'exonuclease trimming and dissected the 21 nt binding motifs. They found that Cbp1 recruits the general chromatin protein Cren7 to the 3'-end of CRISPR repeats, which leads to the suppression of transcription initiation from spacer-internal cryptic promoters, while facilitating transcription from the main promoters in the array leader regions.

The data are supported by several additional approaches. The experiments appear well planned and conducted. The results constitute substantial conceptual advancement and the manuscript is well written.

I have the following comments:

p.2, Introduction: "Chromatinization of DNA regulates transcription in all domains of life, including eukaryotes, bacteria and archaea" Can you please add a suitable citation for this statement, perhaps a more recent review? Especially for bacteria it is relatively unusual to call these processes "chromatinization".

p.5, last paragraph: You mention you tested Cren7 occupancy in strain E234, but 4 lines below, suddenly strain E233S is mentioned. What is the relation between these two strains? Please explain.

p.11: 13,150 TSSs were mapped for the E233S strain and two independent Δ cbp1 strains by Cappable-seq. This number appears very high given the organism has just ~2,400 genes and it seems to be a lot more than in the previous paper Blombach et al. (2021).

Please add more detail here or in the methods section. At present it cannot be reconciled how this number was obtained and to what extent these 13,150 TSSs are statistically sound.

p.20: "While Cbp1 shows some unspecific DNA activity in vitro...." Do you mean ".....DNA-binding activity... "?

p.29, last line in paragraph "Cappable-seq data analysis": delete space in "TSS s".

Reviewer #1 (Remarks to the Author):

CRISPR arrays form an adaptive immune system that necessitates capture and retention of invader sequences. The capture of random, or near random sequences from invaders invokes concerns of capturing sequences that would drive transcription (or other molecular events) from internal spacers within CRISPR arrays. Mechanisms to ensure leader-driven expression of CRISPR arrays to promote adaptive immunity must be balanced with efforts to limit spurious transcription from within arrays. Evidence that nucleoid-binding proteins (NAPs) specifically associate with CRISPR arrays argues that specific NAPs may play critical biological roles in controlling CRISPR expression.

Blombach et al. report on the recruitment and impact of Cbp1 and Cren7 binding to the CRISPR arrays in multiple species of crenarchaeota (the manuscript switches between several species with minimal discussion of the differences in biology of such). In vivo ChIP profiles (from only duplicate samples) form the basis of the primary analyses which are bolstered by in vitro biochemistry and transcription assays. Binding of Cbp1 recruits Cren7 that collectively promotes transcription from leader while limiting internal promoters from firing within CRISPR arrays. However, the impact of Cren7 binding remains unclear given the strong impacts of Cbp1 alone.

The collective data suggests a few main findings that each have some caveats:

1) Cbp1 recruits Cren7 in a sequential manner to bind CRISPR arrays.

Caveats: The authors consistently use the term chromatinization, suggesting a robust level of DNA binding that might approach saturation or invoke a specific nucleoid structure with regulatory potential. However, no data is provided to validate the use of chromatinization as the term is normally applied. Cbp1 and Cren7 do bind the CRISPR arrays, but neither the in vivo nor in vitro data provided imply the formation of a regular structure that would meet the normal definition of chromatinization

We beg to differ with the reviewer, the term chromatin is not strictly defined ('normally') in the literature. We take the view that, in contrast to regulatory transcription factors, chromatin (aka nucleoid-) factors are (i) highly abundant proteins, (ii) able to organise (and in some cases condense) the genome on a larger (kilobase) scale and (iii) have large-scale effects on gene regulation. All these criteria are fulfilled by Cren7 and Cbp1.

The regular structure of histone-based chromatin in eukaryotes was discovered/characterised by nucleosome sequencing (MNase-seq) that generated highly regular pattern formed by arrays of single nucleosomes (higher-order regular structures such as the 30nm fibre are now widely considered to be technical artifacts). Not identical, but similarly, our ChIP-seq and ChIP-exo analyses provide evidence that the patterns of Cbp1 and Cren7 binding to arrays of CRISPR arrays reflects a *highly regular structure*.

There are only few methods to measure absolute levels of chromatinization of DNA such as NoMe-seq or quantitative approaches of MNase-seq (Chereji et al 2019, Genome Biol. 20:198). However, these methods are not protein-specific. The variation of prokaryotic chromatin proteins makes prokaryotic systems so interesting and exciting compared to the 'monoculture' of eukaryotic histones, but, to the best of our knowledge, the diversity of chromatin proteins makes it quasi-impossible to measure directly absolute levels of Cbp1 occupancy.

Our RNA-seq data provide evidence that relative to the genomic region bound by Cbp1 and compared to other, general, chromatin proteins, Cbp1 and Cren7 expression levels should be sufficient to provide high occupancy levels (Supplementary table 2). Moreover, Cren7 is considered a chromatin factor beyond any reproach (e.g., Guo et al. 2008: Nucleic Acids Res 36:1129-37).

Our results show that Cren7 and Cbp1 *jointly* form chromatin structures that regulate gene expression.

To alleviate the reviewer's semantic concerns, we have altered the title to:

'Cbp1-Cren7 form chromatin-like structures on CRISPR arrays that favour transcription from leader-over cryptic promoters'

The image in Figure 7 that shows a near complete saturation of the CRISPR array should be supported by experimental evidence. The ensemble average of billions of genomes through ChIP is not sufficient to define a saturated protein environment due to Cbp1 and Cren7.

Figure 7 is a schematic drawing without any explicit mention of 'complete saturation', neither does the manuscript text, nor figure legends throughout, make that claim. Importantly, nucleosome sequencing likewise does not provide ultimate evidence that 100% of eukaryotic genomes are chromatinised all the time, but nobody would contest that histones are chromatin proteins for that reason.

Also, given that some arrays are not bound by Cbp1 suggests that the general use of arrays should be replaced throughout with Cbp1 bound arrays. The central text of pg 15 highlights this fact, but it is not addressed until page 15.

We have now integrated data from figure 5 (ChIP-seq of Cbp1 in *S. islandicus* LAL14/1) into the first section of the RESULTS section stating earlier that only one family of CRISPR repeats is bound by Cbp1. We do not think, however, that it is helpful to state this distinction explicitly throughout this manuscript. All CRISPR biology is always specific for the cognate repeat sequences in the CRISPR arrays and this is generally not explicitly stated.

2) Cbp1 and Cren7 binding to CRISPR arrays suppresses cryptic transcription while also promoting leader template transcription.

Caveats: While an impact of Cbp1 binding is conclusive with regards to transcription *in vitro*, the manuscript lacks any evidence Cren7 binding influences transcription from leader or internal CRISPR promoters *in vivo* or *in vitro*.

Yes, sure: while it is true that the Cren7 addition does not lead to any profound changes in gene expression, our results show beyond any doubt that both Cren7 and Cbp1 form a mixed chromatin structures *in vivo* and they modulate transcription.

3) A suggestion is made that viral infection differentially regulates both Cbp1 and RNAP recruitment into the CRISPR arrays to impact expression of the array.

Caveats: a true comparison is not available due to contamination issues. Given the failure to be able to correctly analyze the data, this entire section is speculative and uninformative. Either do the experimentation again and provide a proper statistical analysis or remove completely. As written, the authors are making a major claim with massaged and partial data.

Naturally we pay heed to the reviewer's recommendation to improve our manuscript and address the concerns in full in the revised version. However, calling the technical challenges impeding a differential binding analysis "contamination issues" is a mischaracterisation. Background from genomic DNA is always present in ChIP experiments and techniques such as LOESS normalisation have been frequently applied to correct for differences in IP efficiencies that lead to uneven levels of this background. In our

case, the effect is rather strong, but reproducible between the replicates in IPs with different antibodies carried out in parallel with control experiments for uninfected cells. Thus, it is unlikely that repeating the experiment would overcome the lower IP efficiency.

The reviewer is correct that our data do not allow us to quantify such effects. The reviewer might also find that the representation of the data as occupancy plots (IP/input normalised, as is standard) in original Figure 5BC can be perceived as ambiguous as it optically suggests a lower Cbp1 occupancy on CRISPR arrays, a result that cannot be concluded without the use of spike-in controls (even if IP-efficiency would be similar). To improve the data representation, we have removed the occupancy plots for SIRV2-infected cells and added instead a scatter plot that shows MACS2-based enrichment for non-canonical Cbp1 binding sites alongside enrichment in CRISPR arrays 1 and 2 comparing uninfected vs SIRV2-infected cells (Figure 5). The scatter plot shows the absence of larger changes in the relative occupancy of Cbp1 binding sites while clearly reflecting the systematically lower enrichment in the SIRV2-infected cells. We modified the corresponding sentences as follows:

'We used ChIP-seq of Cbp1 and RNA polymerase to specifically test whether virus infection-mediated activation of CRISPR array transcription is accompanied by changes in Cbp1 binding, by comparing uninfected with SIRV2 infected cells. Control RNAP occupancy including increased RNAP occupancy on a type I-A Cas operon consistent with previously published RNA-seq data¹⁰ (Supplementary figure 16). ChIP samples from SIRV-infected cells showed consistently an increased background signal originating from the cell lysate input that somewhat skewed the quantification of Cbp1 occupancy. Despite this, Cbp1 occupancy at non-canonical binding sites appeared to be well correlated between SIRV2-infected and infected cells (Figure 5). When compared to these non-canonical binding sites, Cbp1 occupancy at CRISPR arrays 1 and 2 appeared be unaffected by SIRV2 infection. While we cannot rule out a global reduction in Cbp1 binding upon SIRV2 infection, our data suggest that Cbp1 remains bound to CRISPR arrays when transcription is activated.'

Furthermore, the revised version of accompanying Figure 5 does not any longer compare ChIP-seq occupancy tracks side by side for infected and uninfected cells, aiming to prevent the reader from misinterpretation of the results. Finally, we have revised the statement in the Discussion section on line:

'Our occupancy mapping demonstrates that Cbp1 remains bound to CRISPR arrays during activation of CRISPR systems by SIRV2 infection in LAL14/1 (Figure 5).'

Major concerns require experimental and writing improvements before publication can be endorsed.

Major:

1. The use of multiple species complicates interpretations and is often confusingly written. As a single example, the causal relationship between Cbp1 and Cren7 recruitment into the CRISPR arrays in *S. solfataricus* is inferred using a Cbp1 deletion strain in *S. islandicus*. The authors should minimally include ChIP data of recruitment of both proteins in *S. islandicus* strain. Figure 1d figure title is misleading to use *S. islandicus* WT vs Δ cbp1 when the WT strain is not *S. islandicus* being plotted. Cren7 could potentially be recruited differently in *S. islandicus* compared to *S. solfataricus*. Equating the two different species without very clear clarification is simply not scientifically sound. While this one example is listed, many problems of a similar nature are relevant throughout the manuscript due to the change in species (and changes in making new strains to correct for unexpected deletions).

Working with archaea has its advantages and drawbacks, dependent on the perspective. Researchers using bacterial 'work horses' like *E. coli* or *B. subtilis* are privileged in as much as a wide range of techniques and experimental tractability is available in the same organism. Archaea in contrast, are

much more challenging to work with, and the selection of technique on occasion restricts the choice of model organism, e. g. genetics have been very well developed in *S. islandicus*, much less so in *S. solfataricus*. Be that as it may, this approach offers the advantage of characterising more than one organism, thus providing a broader view of a given biological phenomenon or process. We have now added a paragraph that elaborates on the use of the different species right from the onset on line :

'To gain insight into the chromatinization of CRISPR arrays in vivo and to test whether Cbp1 and Cren7 chromatinization of CRISPR arrays are interdependent, we determined the genome-wide occupancy of Cbp1 by ChIP-seq in three Saccharolobus species: (i) S. solfataricus P2 that harbours six CRISPR arrays (A to F) with slight differences in their repeat sequences, (ii) S. islandicus LAL14/1 that harbours three CRISPR arrays with different, unrelated repeat sequences alongside two CRISPR arrays from the same family of repeats as those found in S. solfataricus 12, and (iii) the genetically tractable S. islandicus REY15A allowing us to test the effect of cpb1 deletion on Cren7 occupancy. S. islandicus and S. solfataricus are closely evolutionary related and the two Cbp1 (P2 vs. REY15A and LAL14/1) proteins share 93% amino acid identity.'

The two species should be classified in the same genus (*Saccharolobus*) and following reviewer 2's suggestion and other recent publications, we use now the term *Saccharolobus islandicus* that reflects biology (diverging from the NCBI taxonomy). Importantly, all our observations were consistent between the two species (and the two *S. islandicus* strains) including the binding site preference and Cbp1 association with CRISPR repeats and *ISC1229* transposons.

2. The use of duplicates only for all ChIP analyses complicates interpretations. The authors must provide some evidence regarding the congruence of the peaks called for replicate #1 and #2. An average of 1 and 99, or 49 and 51, is equal to 50, but the former has a lot of concerns compared to the latter. Inclusion of a simple XY plot of peaks called across the genome in each ChIP experiment for replicate #1 versus replicate #2 would provide more confidence. An R2 value of 0.9 or above would instill confidence in the results. Would the data, and interpretations from such, look nearly identical if only one replicate was used? What, if any, meaningful differences were revealed in the ChIP replicates? Inclusion of a third replicate, with retention and analyses based on using only peaks identified in 2/3 samples would more closely match the norms of publications at this level and vastly improve confidence in the approach presented here.

We agree, and beyond the already provided details on peak calling and filtering regarding reproducibility in the Methods section, we have expanded this section in the revised manuscript with an additional Supplementary Figure 13 that shows scatter plots for the enrichment of peaks unmatched, matched, IDR-filtered, and minimum enrichment-filtered. The R2 value was invariably 0.99 for all Cbp1 ChIP experiments and all subsets of peaks – providing strong evidence of the high degree of reproducibility of these experiments.

3. The absence of a traditional RNA-seq analyses is a major disappointment considering the level of interpretation made regarding recruitment of RNAP and transcription factors to CRISPR arrays that is measured by ChIP. Cappable seq is not a substitute for the impacts that would be revealed by RNA-seq.

We are sorry to learn about the reviewer's major disappointment. And we agree that a global view on changes in gene expression in the *cpb1* deletion strains at the mRNA level has its merit. We have carried out additional RNA-seq experiments and data analysis, however, 'traditional' RNA-seq strategies (as suggested by the reviewer) are not suitable to monitor RNA levels for the *small crRNAs* that are the most likely to be affected by *cpb1* deletion. To overcome this limitation, we have developed and applied a library preparation strategy similar to that used by Quax et al 2013 able to cover both mRNA genes and crRNAs. We added an additional T4 polynucleotide kinase/phosphatase step to remove cyclic 2'-3'

phosphate moieties from crRNAs that improves 3' adaptor ligation. The newly added RNA-seq data reveal two novel interesting aspects: (i) a 5' to 3' polarity effect in CRISPR array transcription as predicted by Martynov et al. (2017), leading to enrichment of newly acquire spacers in the crRNA pool. However, this effect is independent of Cbp1 (Figure 3c), and (ii) Cbp1 binding to CRISPR arrays is required for maintaining transcription levels within the entire chromatin-interaction domain (CID) where the two CRISPR arrays are located. Both of these new insights are very exciting and emphasise the role of Cbp1/Cren7 chromatin on gene regulation in archaea. We report on these new findings in the revised manuscript in a whole new paragraph in the revised version of the manuscript:

'Cpb1 deletion leads to widespread downregulation in the chromosomal environment of CRISPR arrays

To study the effect of cpb1 deletion on transcription more widely beyond cryptic promoters, we carried out RNA-seq using a library preparation strategy designed to include also small RNAs such as crRNAs. The RNA-seq data correlated well between samples for the two WT E233S replicates and for two independent cbp1 strains (Spearman's $r=0.99$ and 1.00 , respectively, Supplementary figure 11). RNA-seq coverage over the CRISPR arrays was dominated by mature crRNAs with the 5' ends of fragments matching the previously described cleavage site for Cas6 endonuclease that processes the precursor transcript into crRNAs²⁰ whereas the 3'-ends showed signs of exonucleolytic trimming (Supplementary figure 12). Notably, crRNA abundance appeared to strongly decline in the 3' half of the CRISPR arrays (Figure 3c) indicating that premature transcription termination might work as a mechanism to enrich crRNAs derived from more recently integrated spacers at the 5' end of CRISPR arrays in species where CRISPR arrays are extensively long⁹. This effect appears to be Cbp1-independent as the cbp1 deletion strains showed the same decline in crRNA abundance towards the 3' half of the CRISPR arrays (Figure 3c).

*To assess the wider effects of cbp1 deletion on transcription, we carried out differential expression analysis for 2655 coding genes with detected expression plus the two CRISPR arrays with 33 and 39 genes significantly up- and downregulated, respectively ($padj < 0.01$). We noted an uneven distribution of significantly downregulated genes with strong clustering around the CRISPR arrays (Figure 3f). Because of this consistent downregulation of transcription in a larger region, we wondered whether this effect could be connected to the chromosome architecture in *S. islandicus* REY15A. Takemata and Bell recently mapped chromosomal interaction domains (CIDs) in *S. islandicus* REY15A using chromosome conformation capture experiments with cells grown under similar growth conditions as in our experiments (exponential growth phase, media supplemented with sucrose and peptide source)²¹. The CRISPR arrays are located within one ~64 kb CID. Significantly downregulated genes were highly enriched within this CID (31 out of 64 genes). Adjusted for predicted operon structures, 18 out 39 predicted operons within this CID contained at least one downregulated gene (Fisher test, adjusted p -value $< 1 \cdot 10^{-26}$, Bonferroni multiple testing correction for all CIDs, up- and down-regulation). This CID-wide downregulation of transcription was also visible in the Cappable-seq data except for the CRISPR array internal TSSs that were upregulated (Supplementary figure 13). Overall, RNA-seq and Cappable-seq corresponded well with 14 genes shared between the 28 and 32 genes significantly differentially regulated, respectively, out of 892 genes shared between both data sets (Fisher's exact test $p < 1 \cdot 10^{-14}$).*

4. The authors suggest the Δ HTH1 and Δ HTH3 variants do not impact expression and thermostability without providing supporting evidence. While the impacts of such deletions could be interpreted as the authors present, alternative interpretations would be partially eliminated by inclusion of additional data. Figure 2c should be accompanied by a statistical analysis between the means of each condition (WT, Δ HTH1, and Δ HTH3), comparing each category (nt change). An ANOVA Pairwise Comparisons for parametric data or Dunn's test for non-parametric data are likely appropriate.

We have included new results in the revised manuscript demonstrating that the thermostability of the mutant proteins at 75 °C is not impaired and similar to the wild type full length factor (Supplementary Figure 5).

While statistical testing is not regularly used for the analysis of biochemical EMSA data, we aim to provide the best analysis of our results. The main aim of these experiments was to test the interaction between DNA template mutations and the Cbp1 HTH deletions regarding their effect on DNA binding. Testing interaction between two variables requires a parametric test, but ANOVA is not well-suited for fraction data such as ours because they are non-negative and bonded (thus violating basic assumptions for the underlying linear regression). Instead, we used beta regression that was specifically developed for fraction data with a log link function. We made the following reasonable assumptions to apply this model: (i) at protein concentrations close to the dissociation constant (~0.5 of DNA bound) the response to a decrease in affinity will be roughly linear, and (ii) the combined effect of protein and DNA mutations (without interaction) would be best modelled as multiplicative (i.e., log link function). Notably, the effect sizes of the interactions are large enough that regardless of the link function and type of regression used for statistical testing, the interaction terms remain highly significant. The updated code is now available in the accompanying github repository. Please note that we also exchanged the WT Cbp1 data for data at 20 nM protein concentration (previously 40 nM) which are closer to the HTH deletion variant data in terms of fraction of DNA bound.

In summary, together with all the other experiments presented in Figure 2, there is only one consistent model for the orientation of Cbp1 binding to the CRISPR repeat and Cren7 recruitment.

5. In vitro transcription assays demonstrate qualitative impacts of Cbp1 and Cbp1/Cren7 DNA binding on transcription from different promoters identified by ChIP binding of transcription components. The absence of any quantification is perplexing and would improve a revised version of the manuscript. The selection of internal sense and internal antisense templates from CRISPR F but the leader template from CRISPR B without proper explanation is perplexing. A competitive experiment of mixing templates with different promoters (leader versus internal) within the same reaction is a clear and obvious addition to the manuscript. Addition of Cbp1/Cren7 under a condition of mixed templates is predicted to differentially impact internal promoters. Furthermore, existing data predict only Cbp1 is necessary to prevent transcription initiation from an internal promoter begging the question of whether or how Cren7 binding may impact repression of internal transcription. The lack of a direct role for Cren7 in vitro, and no evidence of transcription levels in vivo, demands a change of title. Figure 4 also only implicates the activity of Cbp1, not Cren7.

We agree with the reviewer's suggestion, that the template competition experiment can make a valid contribution. The cell-free transcription experiments can be carried out in a single reaction including three different templates modified in such a way that they would produce run-off transcripts of different lengths. But the interpretation of three simultaneously isolated transcripts sets (captured using three different antisense oligonucleotides) in a single affinity purification step would be problematic due to overlapping band patterns of the partial transcripts and longer non-specific transcripts (asterisk in the original Figure 3e). To circumvent this problem, we have repeated the transcription reactions by combining the three different templates in the same reaction and splitting the reactions for separate affinity purification of the specific transcripts. We have furthermore successfully reduced the amount of the long nonspecific transcripts by modifying the 5'-end of the PCR product transcription templates (using amino modifier C12). The new data are included as Supplementary Figure 11.

6. The authors state that "Recruitment of Cren7 could be mediated by direct physical interaction between Cbp1 and Cren7 or by Cbp1-induced topological changes in the DNA template facilitating

Cren7 binding". The authors conclude "Our data thus demonstrate that Cbp1 recruits Cren7 through direct physical interactions". While this may be true, Figure 1f suggests that the presence of DNA is required for Cpb1 and Cren7 interactions. When DNA is not present (first lane after the ladder) or the DNA is non-specific (last lane), the proteins are not interacting. This suggests that DNA is a very important component in these interactions and thus one cannot rule out "Cbp1-induced topological changes in the DNA template facilitating Cren7 binding". Additionally, Figure 2b suggests that DNA is important, as removing Cren7 binding to the DNA abolishes complex formation.

Yes, of course we agree that the interaction is DNA-dependent; this is the reason why we performed the above-mentioned controls in the first place. While our results cannot rule out *additional* Cbp1-induced topological changes helping Cren7 recruitment, our data reveal direct physical interaction between Cbp1 and Cren7 in a DNA-dependent manner. We have now rephrased the last sentence in the paragraph to clarify this:

'Our data thus demonstrate that Cbp1 recruitment of Cren7 depends on direct physical interactions, as well as DNA binding.'

Minor:

The new(er) naming system of the species employed should be used. Minor spelling and grammatical errors persist throughout.

See above. The two species should be classified in the same genus (*Saccharolobus*) and following reviewer 2's suggestion and other recent publications, we use now the term *Saccharolobus islandicus* that reflects biology (diverging from the NCBI taxonomy).

Reviewer #2 (Remarks to the Author):

This manuscript deals with the regulation of CRISPR repeat-spacer array transcription by the CRISPR array binding protein 1 (Cbp1) in Sulfolobales archaea. These arrays are of extraordinary length raising the question how efficient transcription could reach the more distal parts of the arrays.

In this context, two previously made seemingly contrary observations existed: that Cbp1 binds to the CRISPR repeat sequences where it could interfere with transcription, while experimental data showed the opposite, that Cbp1 overexpression actually increased CRISPR-RNA levels.

Technically, the authors mapped Cbp1 binding globally at single-nucleotide resolution by ChIP-seq combined with 5'→3'exonuclease trimming and dissected the 21 nt binding motifs. They found that Cbp1 recruits the general chromatin protein Cren7 to the 3'-end of CRISPR repeats, which leads to the suppression of transcription initiation from spacer-internal cryptic promoters, while facilitating transcription from the main promoters in the array leader regions

The data are supported by several additional approaches. The experiments appear well planned and conducted. The results constitute substantial conceptual advancement and the manuscript is well written.

I have the following comments:

p.2, Introduction: "Chromatinization of DNA regulates transcription in all domains of life, including eukaryotes, bacteria and archaea" Can you please add a suitable citation for this statement, perhaps a more recent review? Especially for bacteria it is relatively unusual to call these processes "chromatinization".

A pubmed search using the query 'bacterial chromatin' yields **2,840 hits**. Recent publications on the topic include:

- Transcription of Bacterial Chromatin., Shen BA, Landick R.J Mol Biol. **2019** Sep 20;431(20):4040-4066. doi: 10.1016/j.jmb.2019.05.041. Epub 2019 May 31.PMID: 31153903
- Determination of the Chromatin Openness in Bacterial Genomes., Al-Bassam MM, Zengler K.Methods Mol Biol. **2023**;2611:63-69. doi: 10.1007/978-1-0716-2899-7_5.PMID: 36807064
- Observing Bacterial Chromatin Protein-DNA Interactions by Combining DNA Flow-Stretching with Single-Molecule Imaging., Kim H, Loparo JJ.Methods Mol Biol. **2018**;1837:277-299. doi: 10.1007/978-1-4939-8675-0_15.PMID: 30109616

A pubmed search using the query 'archaeal chromatin' yields **231 hits**. Recent publications on the topic include:

- The Role of Archaeal Chromatin in Transcription., Sanders TJ, Marshall CJ, Santangelo TJ.J Mol Biol. **2019** Sep 20;431(20):4103-4115. doi: 10.1016/j.jmb.2019.05.006. Epub 2019 May 11.PMID: 31082442
- The interplay between nucleoid organization and transcription in archaealgenomes., Peeters E, Driessen RP, Werner F, Dame RT.Nat Rev Microbiol. 2015 Jun;13(6):333-41. doi: 10.1038/nrmicro3467. Epub **2015** May 6.PMID: 25944489
- Archaea: The Final Frontier of Chromatin., Laursen SP, Bowerman S, Luger K.J Mol Biol. **2021** Mar 19;433(6):166791. doi: 10.1016/j.jmb.2020.166791. Epub 2020 Dec 29.PMID: 33383035

We have added the following references to substantiate the point made on line :

“

p.5, last paragraph: You mention you tested Cren7 occupancy in strain E234, but 4 lines below, suddenly strain E233S is mentioned. What is the relation between these two strains? Please explain.

We apologise for the oversight. E234 has been changed now consistently to E233S. Both designations have been used by the She lab originally.

p.11: 13,150 TSSs were mapped for the E233S strain and two independent Δ cbp1 strains by Cappable-seq. This number appears very high given the organism has just ~2,400 genes and it seems to be a lot more than in the previous paper Blombach et al. (2021). Please add more detail here or in the methods section. At present it cannot be reconciled how this number was obtained and to what extent these 13,150 TSSs are statistically sound.

The TSS data set used in our previous paper was taken from Rotem Sorek's lab (Wurtzel et al. 2010 Genome Research). We specifically applied Cappable-seq here as the most sensitive method for TSS detection to map cryptic promoters throughout the genome. In addition, we list here all TSSs detected, rather than just primary TSSs that we used previously. For comparison, the Wurtzel data set contains additional ~7000 internal sense TSSs as well as a large number of antisense TSSs.

We have added additional information about how our TSS data set compares to Wurtzel et al. in line :

'The Cappable-seq data allowed us to identify 13150 TSSs including 946 primary TSSs for 1506 predicted operons with ~60% of transcripts predicted to be leaderless (5'-UTR length < 4 nt, Supplementary Figure 7) similar to data for S. solfataricus P2¹⁸. The majority of TSSs were internal sense or antisense TSSs (6800 and 4081, respectively) again reflecting findings for S. solfataricus P2¹⁸.'

p.20: "While Cbp1 shows some unspecific DNA activity in vitro...." Do you mean ".....DNA-binding activity... "?

Corrected

p.29, last line in paragraph "Cappable-seq data analysis": delete space in "TSS s".

Corrected

Reviewer #1 (Remarks to the Author):

A thoughtful revision improves the manuscript and publication is warranted. Addition of references for statements in lines 92-103 is warranted.

Reviewer #1 (Remarks on code availability):

Standard GitHub information was available.

Reviewer #2 (Remarks to the Author):

The revised version addresses all my previously made comments. I endorse the publication.

REVIEWERS' COMMENTS

Reviewer #1 (Remarks to the Author):

A thoughtful revision improves the manuscript and publication is warranted. Addition of references for statements in lines 92-103 is warranted.

Reviewer #1 (Remarks on code availability):

Standard GitHub information was available.

We thank the reviewer for their time and input helping us to improve our manuscript We added the additional references as follows:

New spacers that provide greater protection against the current viral pool are generally integrated at the 5' end of CRISPR arrays 12-15. Premature transcription termination enriches the crRNAs bearing new spacers in the total crRNA pool and counteract the 'dilution effect' arising from the transcription of distal spacers in very long CRISPR arrays 11.

References:

- 11 Martynov, A., Severinov, K. & Ispolatov, I. Optimal number of spacers in CRISPR arrays. *PLoS Comput Biol* 13, e1005891, doi:10.1371/journal.pcbi.1005891 (2017).
- 12 Rollie, C., Schneider, S., Brinkmann, A. S., Bolt, E. L. & White, M. F. Intrinsic sequence specificity of the Cas1 integrase directs new spacer acquisition. *Elife* 4, doi:10.7554/eLife.08716 (2015).
- 13 Barrangou, R. et al. CRISPR provides acquired resistance against viruses in prokaryotes. *Science* 315, 1709-1712, doi:10.1126/science.1138140 (2007).
- 14 Swarts, D. C., Mosterd, C., van Passel, M. W. & Brouns, S. J. CRISPR interference directs strand specific spacer acquisition. *PLoS One* 7, e35888, doi:10.1371/journal.pone.0035888 (2012).
- 15 Yosef, I., Goren, M. G. & Qimron, U. Proteins and DNA elements essential for the CRISPR adaptation process in *Escherichia coli*. *Nucleic Acids Res* 40, 5569-5576, doi:10.1093/nar/gks216 (2012).

Reviewer #2 (Remarks to the Author):

The revised version addresses all my previously made comments. I endorse the publication.

We would like to thank the reviewer for their valuable input.